# DIFFERENTIABLE *and* TRANSPORTABLE STRUCTURE LEARNING

## ABSTRACT

Directed acyclic graphs (DAGs) encode a lot of information about a particular distribution in its structure. However, compute required to infer these structures is typically super-exponential in the number of variables, as inference requires a sweep of a combinatorially large space of potential structures. That is, until recent advances made it possible to search this space using a differentiable metric, drastically reducing search time. While this technique— named NOTEARS —is widely considered a seminal work in DAG-discovery, it concedes an important property in favour of differentiability: *transportability*. To be transportable, the structures discovered on one dataset must apply to another dataset from the same domain. In our paper, we introduce *D-Struct* which recovers transportability in the discovered structures through a novel architecture and loss function, while remaining completely differentiable. Because D-Struct remains differentiable, our method can be easily adopted in existing differentiable architectures, as was previously done with NOTEARS. In our experiments, we empirically validate D-Struct with respect to edge accuracy and structural Hamming distance in a variety of settings.

## 1 INTRODUCTION

Machine learning has proven to be a crucial tool in many disciplines. With successes in medicine [1–5], economics [6–8], physics [9–14], robotics [15–18], and even entertainment [19–21], machine learning is transforming the way in which experts interact with their field. These successes are in large part due to increasing accuracy of diagnoses, marketing campaigns, analyses of experiments, and so forth. However, machine learning has much more to offer than improved accuracy alone. Indeed, recent advances seem to support this claim, as machine learning is slowly recognised as a tool for scientific discovery [22–25]. In these successes, machine learning helped to uncover a previously unknown relationships between variables. Discovering such relationships is the *first step* of the long process of scientific discovery and are the focus of our paper as D-Struct— the model we propose in this paper —aims to help through differentiable *and* transportable structure learning.

**The structures.** We focus on discovering *directed acyclic graphs* (DAGs) in a domain $\mathcal{X}$. A DAG helps us understand how different variables in $\mathcal{X}$ interact with each other. Consider a three-variable domain $\mathcal{X} := \{X, Y, Z\}$, governed by a joint-distribution, $\mathbb{P}_{\mathcal{X}}$. A DAG explicitly models variable interactions in $\mathbb{P}_{\mathcal{X}}$. For example, consider the following DAG: $\mathcal{G} = X \longrightarrow Z \longrightarrow Y$, where $\mathcal{G}$ depicts $\mathbb{P}_{\mathcal{X}}$ as a DAG. Such a DAG allows useful analysis of dependence and independence of variables in $\mathbb{P}_{\mathcal{X}}$ [26, 27]. From $\mathcal{G}$, we learn that $X$ does not directly influence $Y$, and that $X \perp\!\!\!\perp Y | Z$ as $X$ does not give us any additional information on $Y$ once we know $Z$. While DAGs are the model of choice in causality [28], it is impossible to discover a *causal* DAG from observational data alone [29–32]. As we only wish to assume access to observational data, our goal is not *causal* discovery.

The above forms the basis for conventional DAG-structure learning [33]. In particular, $X \perp\!\!\!\perp Y | Z$ strongly limits the possible DAGs that model $\mathbb{P}_{\mathcal{X}}$. Given more independence statements, we limit the potential DAGs further. However, independence tests are computationally expensive which is problematic as the number of potential DAGs increases super-exponentially in $|\mathcal{X}|$ [34].

This limitation strongly impacted the adoption of DAG-learning, until Zheng et al. [35] proposed NOTEARS which incorporates a differentiable metric to evaluate whether or not a discovered structure is a DAG [35, 36]. Using automatic differentiation, NOTEARS learns a DAG-structure in a much more efficient way than earlier methods based on conditional independence tests (CITs).

Figure 1: **Transportability in DAG discovery.** Different patients go to different hospitals (left), yet we wish to infer a *general* structure (right) *across* hospitals. A structure can only be considered a discovery if it generalizes in distributions over the same domain. For example, the way blood pressure interacts with heart disease is the same for all humans and should be reflected in the structure.

While NOTEARS makes DAG inference tractable, we recognise an important limitation in the approach: a discovered DAG does not generalise to equally factorisable distributions, i.e. NOTEARS is not *transportable*. While we explain why this is the case in Section 2.2 (and confirm it empirically in Section 4), we give a brief description of the problem below, helping us to state our contribution.

**Transportability.** Consider Fig. 1, depicting two hospitals: 🏥 and 🏥, named hospitals A and B onward. Each hospital hosts patients described by the same set of features such as age and gender. However, the hospitals may have different patient-distributions, e.g. patients in A are older compared to B. *But their underlying biology remains the same.* Using NOTEARS to learn a DAG from data on hospital A, actually does not guarantee the same DAG is discovered from data in hospital B.

Learning from multiple data-sources is not new. In particular, papers focusing on *federated structure learning* solve a similar objective as described above [37, 38]. However, we believe transportability is a more general property than only training from multiple data sources. Crucially, transportability is very explicit about the domains we learn from, allowing their distributions to *vary* across domains.

Interestingly, despite computational limitations, transportability is actually guaranteed when using a CIT-based discovery method [39, 40], assuming that patients in both hospitals exhibit the same (in)dependencies in $\mathcal{X}$. Being unable to transport findings across distributions is a major shortcoming, as replicating a found discovery is considered a hallmark of the scientific method [41–44].

**Contributions.** In this paper, we present *D-Struct*, the first *transportable* differentiable structure learner. Transportability grants D-Struct several advantages over the state-of-the art: D-Struct is more robust, even in the conventional single-dataset case (Table 1); D-Struct is fast, in fact, we report time-to-convergence often up to 20 times faster than NOTEARS (Fig. 5); and given its completely differentiable architecture, D-Struct is easily incorporated in existing architectures (e.g. [45–49]).

While transportable methods have clear benefits over non-transportable methods in settings with multiple datasets (as illustrated in Fig. 1), we emphasise that *our method is not limited to these settings alone*. In fact, we find that enforcing transportability significantly increases performance in settings with one dataset, which is arguably most common. In Section 3 we introduce D-Struct and how to use our ideas in the single dataset setting. We then empirically validate D-Struct in Section 4.

## 2 PRELIMINARIES AND RELATED WORK

Our goal is to build a transportable *and* differentiable DAG learner. Without loss of generality, we focus our discussion mostly on NOTEARS [35] (and refinements [36, 50–52]) as it is the most adopted differentiable DAG learner. For a more in depth overview of structure learners (CIT-based as well as score-based), we refer to Appendix G or relevant literature [26, 28, 34]. First, we formally introduce transportability, and then explain how NOTEARS works and why it is not transportable.

### 2.1 TRANSPORTABILITY

**Factorisation and independence.** Consider a distribution, $\mathbb{P}_\mathcal{X}$, which we can factorise into,

$$\prod_i \mathbb{P}_{\mathcal{X}_i|\mathcal{X}_{i+1:d}}, \tag{1}$$

with $i \in [d]$, where $[d] := 1, \ldots, d$, and $\mathcal{X}_i$ representing the $i^{\text{th}}$ element in $\mathcal{X}$. Eq. (1) may get quite long with increasing $d$ as the conditions may contain up to $d-1$ different variables. This becomes

restrictive when estimating and decomposing $\hat{\mathbb{P}}_{\mathcal{X}}$ from data. Instead, we can simplify eq. (1) using independence statements, e.g. $\mathcal{X}_i \perp\!\!\!\perp \mathcal{X}_k$ invokes the equality: $\mathbb{P}_{\mathcal{X}_i | \mathcal{X}_{j,k}} = \mathbb{P}_{\mathcal{X}_i | \mathcal{X}_j}$. A simplified version of eq. (1) translates into a smaller *Markov blanket and boundary* [53] (see Appendix D).

**Non-symmetrical statements.** Contrasting the above, we are interested in *directed* and *acyclic* graphical (DAG) structures. Let $\mathcal{G}_{\mathcal{X}} \coloneqq \{\mathcal{X}, \mathcal{E}\}$ be a DAG, where $\mathcal{E} \subset \mathcal{X} \times \mathcal{X}$ is a set of edges connecting nodes in $\mathcal{X}$ (i.e. random variables), with $(\mathcal{X}_i, \mathcal{X}_j) \in \mathcal{E}$ implying $(\mathcal{X}_j, \mathcal{X}_i) \notin \mathcal{E}$ [54].

While independence is symmetric, it is still possible to infer non-symmetric structures with only independence statements using d-separation [28, 30, 55–58]. Given a collection of conditional independence statements, which renders two variables independent given a third, e.g. $X \perp\!\!\!\perp Y | Z$, d-separation (defined in Def. 4 in Appendix D) helps us narrow down a directed structure from independence statements [59, 60][1]. If a set $\mathcal{X}_d$ blocks all paths between $\mathcal{A}$ and $\mathcal{B}$, they are d-separated, d-sep$_{\mathcal{G}}(\mathcal{A}; \mathcal{B} | \mathcal{X}_d)$, where blocking is directly related to independence between $\mathcal{A}$ and $\mathcal{B}$.

With d-separation and the common faithfulness assumption (see Appendix G), we have a link between $\mathcal{G}_{\mathcal{X}}$ and $\mathbb{P}_{\mathcal{X}}$. Specifically, the conditional independence relationships in $\mathcal{G}_{\mathcal{X}}$ have a one-to-one correspondence with those in $\mathbb{P}_{\mathcal{X}}$ [61], i.e. if $X \perp\!\!\!\perp_{\mathbb{P}} Y | Z$ then $X \perp\!\!\!\perp_{\mathcal{G}} Y | Z$, where $\perp\!\!\!\perp_{\mathcal{S}}$ denotes independence in $\mathcal{S}$. The reverse is not necessarily true as there can be many graphs that correspond with $\mathbb{P}$ in terms of (in)dependence— these graphs are termed Markov equivalent [26].

The set of conditional independence assertions in $\mathbb{P}$ is denoted as $\mathcal{I}(\mathbb{P})$. Similarly, all independence statements implied by d-separation in a graph $\mathcal{G}$ are denoted as $\mathcal{I}(\mathcal{G}) = \{(\mathcal{X} \perp\!\!\!\perp \mathcal{B} | \mathcal{X}_d) : \text{d-sep}_{\mathcal{G}}(\mathcal{A}; \mathcal{B} | \mathcal{X}_d)\}$, referred to as the set of *global Markov independencies* [26, Chapter 3].

**Invariance and discovery.** Consider two datasets: $\mathcal{D}_1 = \{X^{(n)} \in \mathcal{X} : n \in [N]\}$ and $\mathcal{D}_2 = \{X^{(m)} \in \mathcal{X} : n \in [M]\}$, spanning the same space $\mathcal{X}$. As a sample $X^{(n)}$ from $\mathcal{D}_1$ depicts the same variables as a sample $X^{(m)}$ from $\mathcal{D}_2$, both datasets reflect the *same* underlying mechanisms. For example, if hospital A collected a dataset on its patients in $\mathcal{X}$ (say $\mathcal{D}_1$) and associated smoking with cancer, then if this statement is true this should also be found in data collected by hospital B ($\mathcal{D}_2$).

Of course, while the samples in $\mathcal{D}_1$ and $\mathcal{D}_2$ come from the same domain $\mathcal{X}$, they may be sampled from different distributions, $\mathbb{P}^1_{\mathcal{X}}$ and $\mathbb{P}^2_{\mathcal{X}}$, respectively. Recall Fig. 1, where hospitals A and B may be located in different regions, resulting in the characteristics of patients to also be different. However, key in a scientific discovery is that it generalises *beyond* distributions and is carried over through the entire domain $\mathcal{X}$. In other words, any structure we may find in $\mathcal{D}_1$, should also be found in $\mathcal{D}_2$, as for almost all distributions $\mathbb{P}^i_{\mathcal{X}} \in \mathcal{P}$ that factorise over $\mathcal{G}^2$, $\mathcal{I}(\mathbb{P}^i_{\mathcal{X}}) = \mathcal{I}(\mathcal{G}) = \mathcal{I}(\mathbb{P}^j_{\mathcal{X}})$ where $\mathbb{P}^i_{\mathcal{X}} \neq \mathbb{P}^j_{\mathcal{X}}$ [26, Theorem 3.5]; if this is not the case, we haven't discovered anything at all.

**Definition 1** (Transportability). *With multiple datasets, $\{\mathcal{D}_k \sim \mathbb{P}^k_{\mathcal{X}} : k \in [K]\}$ over the same domain $\mathcal{X}$, sampled from potentially different distributions $\mathbb{P}^i_{\mathcal{X}} \neq \mathbb{P}^j_{\mathcal{X}}$ if $i \neq j$ for all $i, j \in [K]$, we call a method transportable if it learns a structure that is the same across all datasets: $\{\mathcal{D}_k \to \mathcal{G}_k : k \in [K]\}$ s.t. $\mathcal{G}_1 = \cdots = \mathcal{G}_K$.*

Def. 1 states that when a DAG found in $\mathcal{D}_1$ is also found in $\mathcal{D}_2$, we consider that DAG, and the method proposing it, *transportable*. Transportability in CIT-based methods is only satisfied when we assume that each distribution contains the same global Markov independencies. This assumption is not strict at all, as we are only concerned with distributions that span the same domain $\mathcal{X}$, and thus inherit the same interactions as posed by $\mathcal{X}$, i.e. the discoveries to be made. As such, CIT-based methods learn transportable DAGs automatically as transportability is a property directly related to the set of independencies of both distributions and DAGs, which we assume to be the same.

From the above, it is clear that transportability is a property of the structure learning method itself. In the case of CIT-based methods, we are guaranteed transportability in our setting, but not so for differentiable structure learnings. However, given that the method is responsible for transportability, we can come up with ways to include this property in differentiable structure learners also.

---

[1]Note that in these earlier works, DAGs were named *influence diagrams*.

[2]For all distributions except for a set of measure zero in the space of conditional probability distribution parameterizations [39].

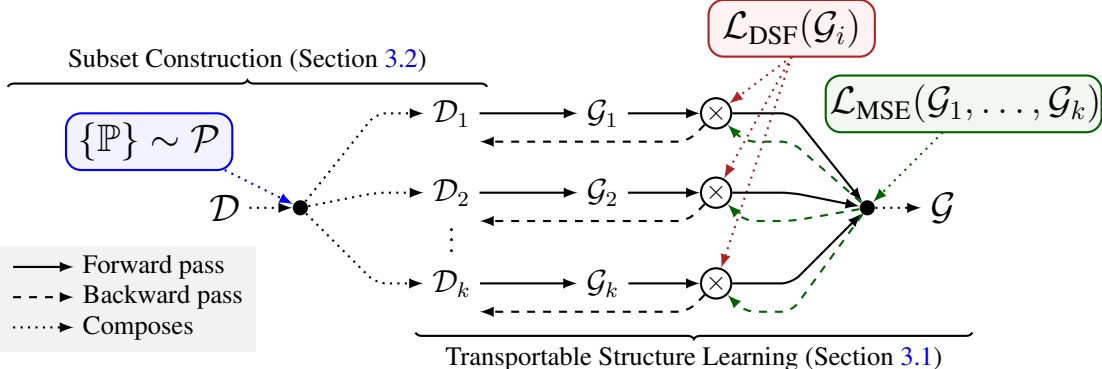

Figure 2: **D-Struct architecture.** Above architecture enforces transportability. D-Struct is split in two major parts: subset construction (Section 3.2) and the transportable structure learning algorithm (Section 3.1). There are three components: $\mathcal{P}$, $\mathcal{L}_{\text{DSF}}$, and $\mathcal{L}_{\text{MSE}}$. The losses are combined and backpropagated through the architecture. Lastly, all DSFs are merged into a final DAG structure $\mathcal{G}$.

## 2.2 DIFFERENTIABLE STRUCTURE LEARNING

CIT-based methods evaluate each possible DAG that spans $\mathcal{X}$ with respect to its factorisation over $\mathbb{P}_{\mathcal{X}}$ using $\mathcal{I}(\mathcal{G} \in \mathbb{G}_{\mathcal{X}})$, where $\mathbb{G}_{\mathcal{X}}$ denotes the space of all possible DAGs in the domain $\mathcal{X}$. The major issue with this is computation. Essentially, there are two aspects which negatively impact computation time: first, the number of to-be-evaluated DAGs in $\mathbb{G}_{\mathcal{X}}$ increases super-exponentially in $|\mathcal{X}|$ (e.g. 10 variables result in $> 4 \times 10^{18}$ possible DAGs [34, 62, 63]); second, simply recovering $\mathcal{I}(\mathbb{P}_{\mathcal{X}})$ to evaluate each $\mathcal{G} \in \mathbb{G}_{\mathcal{X}}$ requires many independence tests, each with additional compute. Appendix G includes an overview of the most well known CIT-based (and score-based) methods.

**Differentiable score functions.** Enter *differentiable score functions* (DSFs). With DSFs one *can* traverse $\mathbb{G}_{\mathcal{X}}$ smartly to arrive at a DAG [64, 65] much faster [63]. Furthermore, a differentiable method is straightforward to include in a variety of differentiable architectures, allowing joint optimisation of both the graphical structure as well as the accompanying structural equations or another downstream use.

Most notable is NOTEARS by Zheng et al. [35], which proposes the following learning objective:

$$\min_{A \in \mathcal{A}} F(A) + \lambda_1 \|A\|_1 + \frac{\rho}{2}|h(A)|^2 + \lambda_2 h(A), \tag{2}$$

where $A \in \mathbb{R}^{d \times d}$ (transformed to binary with a threshold) denotes an adjacency matrix; $F(A)$ is a likelihood based loss (like the MSE); $\rho$ and $\lambda_{1,2}$ are parameters of the augmented Lagrangian; and

$$h(A) := \text{tr}(\exp(A \circ A)) - d, \tag{3}$$

is the actual differentiable score function, where $\text{tr}(\cdot)$ is the matrix trace operator, and $\circ$ is the element-wise (Hadamard) product. Importantly, $h(A) = 0$ indicates $A$ is a DAG. Considering that eq. (3) is differentiable, we can take its derivative with respect to $A$ and minimise eqs. (2) and (3).

As is always the case for gradient-based learning, the random initialisation of $A$ may guide the optimisation in different directions, potentially arriving at a local minimum in the case of non-convex targets. The latter is certainly the case in more recent improvements of NOTEARS as they almost exclusively focus on non-linear structural equations which result in non-convex losses [36, 51, 52].

**Transportability of DSFs.** Current DSFs are not transportable, due to eq. (2) having conflicting solutions— contrasting the single solution (set) that transforms $\mathcal{I}(\mathbb{P})$ to $\mathcal{G}$. Essentially, the approximate nature of (stochastic) gradient-based learning can result in conflicting estimate structures given different datasets. In fact, other works recognise this [66]. Rather than trying to recover one true DAG, their objective is to learn multiple (potentially different DAGs) through a multi-task objective. In our work, we assume there *does* exist one (unique) DAG, which is the one we aspire to recover.

## 3 D-STRUCT: *DIFFERENTIABLE AND TRANSPORTABLE Struct*URE LEARNING

The goal of any structure learner (differentiable or not) is to transform finite data into a structure:

$$\mathcal{D} \to \mathcal{G},$$

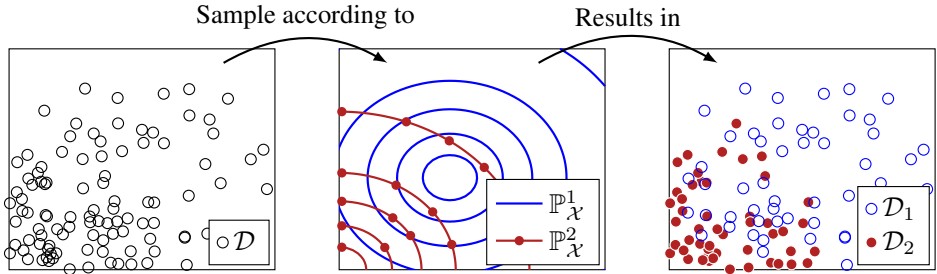

(a) We are presented with a dataset $\mathcal{D}$ over the domain $\mathcal{X}$.

(b) With two distributions $\mathbb{P}^1_{\mathcal{X}}$ and $\mathbb{P}^2_{\mathcal{X}}$, we can sample from $\mathcal{D}$.

(c) Sampling according to the two distributions results in two subsets $\mathcal{D}_1 \cup \mathcal{D}_2 = \mathcal{D}$.

Figure 3: **Differently distributed *single-origin* data.** On the left we illustrate a single-origin dataset $\mathcal{D}$, sampled from one distribution. In the middle we illustrate two distributions over the domain of $\mathcal{D}$, which are used to resample two subsets from $\mathcal{D}$, thereby creating a new multi-origin datasource.

and so is the case for D-Struct. We introduce D-Struct in Section 3.1 and provide implementation details using NOTEARS in Section 3.3. While transportability seems most natural in a setting with multiple datasets (e.g. data from two hospitals illustrated in Fig. 1), we show in Section 3.2 how one can leverage transportability in the setting where there is only one dataset. In fact, our experiments in Section 4 show that D-Struct consistently offers more accurate DAGs in this "single-origin" setting.

### 3.1 D-STRUCT: TRANSPORTABLE STRUCTURE LEARNING

In order to enforce transportability, D-Struct employs an ensemble-architecture of multiple initialisations of a chosen DSF and their appropriate (differentiable) architecture. Each loss is then combined with a regularisation function based on the D-Struct architecture. Fig. 2 depicts this architecture and highlights how our regularisation scheme is backpropagated throughout the entire network.

Given datasets $\mathcal{D}_1, \ldots, \mathcal{D}_K$, we can use any DSF (e.g. [35, 36, 50–52]) to learn a DAG. Specifically, we let $K$ *distinct* DSFs learn a DAG from one of the $K$ datasets, agnostic from each other. We consider these learning objectives to be $K$ parallel objectives, as is also illustrated in Fig. 2 in the rightmost part. Crucially, D-Struct does not restrict which type of DSF we can use. In a linear setting one can use vanilla NOTEARS [35], whereas in a non-linear setting, one can use the non-parametric version [36]. Naturally, any restriction posed by the chosen DSF will carryover to D-Struct. We use NOTEARS-MLP [36] in Sections 3.3 and 4, while Appendix A includes pairings with other DSFs.

At this point, we identify a first loss term: $\mathcal{L}_{\text{DSF}}(\mathcal{G}_k)$, which depends on the chosen DSF (illustrated in red in Fig. 2). In the case of NOTEARS, $\mathcal{L}_{\text{DSF}}(\mathcal{G}_k)$ corresponds with eqs. (2) and (3). Whenever data is passed through the architecture– without mixing distinct datasets –we evaluate the discovered structure as $\mathcal{L}_{\text{DSF}}(\mathcal{G}_k | \mathbf{X} \sim \mathcal{D}_k)$, where $\mathbf{X} \subseteq \mathcal{D}$. If the chosen DSF requires hyperparameters (such as $\lambda_{1,2}$ and $\rho$ in eq. (4)), we have to also include these in D-Struct's set of required hyperparameters. While it is possible to set different hyperparameter values for each of the DSFs separately (which is potentially helpful when there is a lot of variety in the $K$ distinct datasets), we fix these across DSFs in light of simplicity. A discussion on D-Struct's hyperparameters can be found in Appendix A.1.

Given $\{\mathcal{L}_{\text{DSF}}(\mathcal{G}_k) : k \in [K]\}$ we enforce transportability across each $\mathcal{D}_k$ by comparing the structures $\mathcal{G}_1, \ldots, \mathcal{G}_k$. We do this by calculating the difference of the adjacency matrices $A_k \in \mathbb{R}^{d \times d}$. Specifically, for each gradient calculation (before we perform a backward pass), we take the (element-wise) mean adjacency matrix, $\bar{A}_{1:K} = \frac{1}{K} \sum_k A_k$, detach it from the gradient and backpropagate the MSE for each parallel DSF. In particular, we include the following regularisation term in D-Struct's loss:

$$\mathcal{L}_{\text{MSE}}(A_k) := \|A_k - \bar{A}_{1:K}\|_2^2. \tag{4}$$

Minimising eq. (4) results in transportable structures (see Theorem 1). Note that eq. (4) (green in Fig. 2) remains differentiable, which was our goal for D-Struct. We add $\mathcal{L}_{\text{MSE}}(\mathcal{G}_k)$ to the DSF loss,

$$\mathcal{L}(\mathcal{G}_k | \mathcal{D}_k) := \mathcal{L}_{\text{DSF}}(\mathcal{G} | \mathcal{D}_k) + \alpha \mathcal{L}_{\text{MSE}}(A(\mathcal{G}_k)), \tag{5}$$

where $A(\mathcal{G})$ indicates the adjacency matrix of $\mathcal{G}$, and $\alpha$ is a scalar hyperparameter (refer to Appendix A.1 for hyperparameter settings, details, and further insights). Note that the second term in eq. (5) does not depend on $\mathcal{D}_k$. Having $\mathcal{L}_{\text{MSE}}$ be agnostic to the data makes sense as transportability

is not a property of the data. Indeed, recall from Section 2.1 that transportability is a property of the structure learner instead. This additional term (based solely on the parallel learners) enforces transportability as the architecture encourages the DSFs to converge to the same adjacency matrix.

**Theorem 1** (**Minimising eq. (4) yields transportable structures.**). *The only way in which eq. (4) is equal to 0— for **every** adjacency matrix $A_k$ —is when $A_1 = \cdots = A_K$. Even a slight difference in one of the $A_k$ will result in a non-zero equation 4 as $\bar{A}_{1:K}$ will be affected by this difference. Having every $A_1 = \cdots = A_K$ and thus equal structures in $\mathcal{G}_k$— where each $A_k$ is learned from a distinct $\mathcal{D}_k$ —corresponds with transportable structures as we have defined in Def. 1* □

## 3.2 D-STRUCT: SUBSET CONSTRUCTION

So far, we have only discussed the rightmost part of Fig. 2. In Section 3.1, we assumed data is provided in multiple distinct datasets, i.e. they stem from a multi-origin datasource. However, here we explain how even in the single-origin case D-Struct is applicable, irrespective of which DSF we end up choosing. As such, we continue our discussion by explaining the leftmost part of Fig. 2. Naturally, if one already has distinct $\mathcal{D}_k \sim \mathbb{P}^k$, this step can be skipped and D-Struct can be used as in Section 3.1.

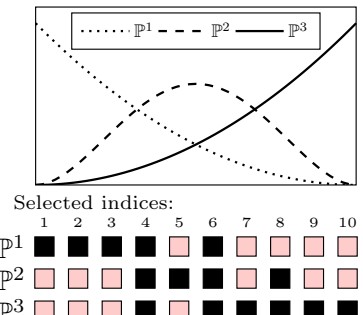

Selected indices:

Figure 4: $K$ **distributions.** We have illustrated the subset sampling with beta-distributions above, for $K = 3$. For each density, we evaluate its PDF for each index, normalize it and perform a Bernoulli experiment. The selected indices are plotted below the density functions (black indicates a selected index).

Different distributions will guide each (distinct) optimisation target into a different direction. Combining their results, will encourage the total model to be more robust and generalisable. However, while a multi-origin datasource may be governed by multiple distributions, a single-origin one is not. Our task is clear: from a single-origin datasource, we have to *mimic* a multi-origin datasource in such a way that we know each subset has a different distribution, yet maintains the properties of the original single-origin-distribution (such as the global Markov independencies). Doing so allows us to enforce transportability through eq. (4).

The lefthand side of Fig. 2 shows that we handle the single-origin case by transforming it to a multi-origin case. With this *constructed* multi-origin setup, we can continue using D-Struct as we have done in Section 3.1. We preface the multi-origin case with a step that divides $\mathcal{D}$ into subsets $\{\mathcal{D}_1, \ldots, \mathcal{D}_k\}$, according to different distributions $\mathcal{P} := \{\mathbb{P}^1_{\mathcal{X}}, \ldots, \mathbb{P}^k_{\mathcal{X}}\}$. Consider Fig. 3 where we illustrate how we sample from $\mathcal{D}$ using $\mathbb{P}^k \in \mathcal{P}$. In principle, each element $X^{(n)} \in \mathcal{D}$ has a $\mathbb{P}^k(X^{(n)})$ probability to be sampled from $\mathcal{D}$, for each $\mathbb{P}^k \in \mathcal{P}$. As such, each distribution leads to a subset $\mathbb{P}^k \times \mathcal{D} \to \mathcal{D}_k$ where $\bigcup_k \mathcal{D}_k = \mathcal{D}$, and $\mathcal{D}_k$ need not be disjoint but is not equal to $\mathcal{D}$.

In our experiments, we perform this preprocessing step by first correlating the index of each element in $\mathcal{D}$ with their values in $\mathcal{X}$. Next, we define $K$ distributions over $[N]$ and then use these distributions to sample indices. The sampled indices compose the subset. While we have included a detailed description of our implementation in Appendix F, we give a brief step-wise explanation below.

◆ **Step 1** *Correlating indices and values.* Sorting and reindexing elements in $\mathcal{D}$ according to the covariates in $\mathcal{X}$ ensures a dependency between $\mathcal{X}$ and $i \in [N]$, where $i < j$ indicates $X^{(i)} < X^{(j)}$, i.e. the *order* of $X$'s in the data structure representing $\mathcal{D}$ is correlated with the *values* of the $X$'s.

◆ **Step 2** *Distributions over $[N]$.* Step 1 allows us to create subsets based on one-dimensional distributions $\{\mathbb{P}^k_{[N]} : k \in [K]\}$, rather than much more complicated distributions over $\mathcal{X}$. An added bonus to these one-dimensional distributions is that they easily scale to more dimensions in $\mathcal{X}$. Of course, the number of distributions, and consequentially their shape, should change in function of $K$. Specifically, with higher $K$, we have to ensure that the probability mass of each distribution is concentrated in different areas of $[N]$. As such, we chose to model these as beta-distributions with,

$$\alpha, \beta \in \{(i, K), (K, K), (K, j) : i \in \text{interp}(1, K - 1), j \in \text{interp}(K - 1, 1)\},$$

where $\text{interp}(a, b)$ is a linear interpolation between $a$ and $b$, used to sample $\lfloor \frac{K}{2} \rfloor$ $i$'s and $j$'s. When $K$ is even we leave out $(K, K)$ such that the total number of distributions always equals $K$.

◆ **Step 3** *Selecting indices.* Our final task is to create $K$ subsets, which due to the first step is

simplified to choosing indices instead. These indices are selected based on the distributions defined in the second step. First, we evaluate each density's PDF for every index (after normalisation: $\frac{i}{N}$), and normalise the output to be a value between $0$ and $1$. Once we have $K$ values for each index, we perform a Bernoulli experiment where the output determines whether or not the index is selected to be part of the subset $k \in [K]$. This process is illustrated in Fig. 4 for $K = 3$ using beta distributions.

With each subset, we apply D-Struct as explained in Section 3.1. In our experiments (Section 4 and Appendix A), we show that D-Struct greatly improves performance of non-transportable DSFs. Furthermore, we also empirically validate our subsampling routing compared to random sampling.

### 3.3 EXAMPLE IMPLEMENTATION USING NOTEARS-MLP

D-Struct works with any DSF, though it is instructive to illustrate this with an example. For this, we chose NOTEARS-MLP [36] which is a non-parametric (cfr. the structural equations) extension of the classic NOTEARS paper [35]. The main challenge to incorporating D-Struct into NOTEARS-MLP is to integrate it into its dual ascent strategy which solves the (non-convex) constrained optimisation problem in eq. (2) [50], solved with an augmented Lagrangian method [67, Chapter 5].

The constraint in the optimisation problem stems from, for example, knowing that the diagonal of $A$ can only contain zeros [35, 36, 50]. NOTEARS (and its extensions), solve this problem by using the L-BFGS-B optimizer [68] which can handle parameter bounds out-of-the-box, making it a suitable choice to optimise the augmented Lagrangian[3]. This is made explicit in Algs. 1 and 2.

**Init.:** $\theta_k$ for each $k \in [K]$
**Input:** $h_{\text{tol}}, \rho_{\max}$
**Setup:** $h \leftarrow \infty, \rho_{1,\ldots,K} \leftarrow 1, \rho \leftarrow 1$
**for** *maximum amount of epochs* **do**
    **for** $k \in [K]$ **do**
        **for** *batch* $\sim \mathcal{D}_k$ **do**
            `training_step`$(\theta_k,$
             `batch`$)$;
            $h \leftarrow \max_k h(A(\theta_k))$;
            $\rho \leftarrow \min_k \rho_k$;

**Algorithm 1:** Outer-loop of dual ascent procedure for D-Struct(NOTEARS-MLP)

**Input:** $\theta_k$, batch
**while** $\rho < \rho_{\max}$ **do**
    $l_{\text{m}} \leftarrow \mathcal{L}_{\text{MSE}}(\theta_1, \ldots, \theta_K)$;
    $l_{\text{d}} \leftarrow \mathcal{L}_{\text{DSF}}(batch)$;
    $\theta \leftarrow$ `L-BFGS-B.update`$(l_{\text{m}}, l_{\text{d}})$;

    $h' \leftarrow h(A(\theta_k))$;
    **if** $h' > 0.25h$ **then**
        $\rho_k \leftarrow 10\rho_k$;
    **else**
        `break`;

**Algorithm 2:** `training_step` for D-Struct(NOTEARS-MLP)

Algorithms 1 and 2 highlight where D-Struct requires edits in the standard NOTEARS-MLP algorithm. Most obvious is the creation of multiple parameters, $\theta_k$ for each $k \in [K]$, where each $\theta_k$ indicates the set of parameters for one initialisation of NOTEARS-MLP, exactly following the architecture depicted in Fig. 2. The set $\{\theta_k : k \in [K]\}$ then denotes the parameters for D-Struct. As such, the number of parameters for D-Struct scales linearly in $K$, compared to the used DSFs.

From Algs. 1 and 2, we learn that information across the different NOTEARS-MLPs is shared in `training_step` (corresponding with Alg. 2). Typically, a training step is solely focused on one structure learner leaving the learner unaware of the other DSFs, as is also implied in Alg. 1 which iterates over each learner separately. The sharing of information *across* each learner though— through $\mathcal{L}_{\text{MSE}}(\theta_1, \ldots, \theta_k)$ computed in the first line in Alg. 2's while loop —is enforcing transportability.

D-Struct hardly increases implementation complexity. In fact, besides architectural alterations (as explained in Section 3.1 and Fig. 2), the optimisation strategy is mostly adopted from the underlying DSF. This is an important advantage. In fact, Zheng et al. [35] already state the importance of an easy to implement model; they use approximately 60 lines of code, and we only add approximately 10 lines to that. Furthermore, we also noticed a major improvement in efficiency as D-Struct drastically reduces computation time compared to NOTEARS. We report computation time in the next section.

---

[3]This also allows including prior knowledge on $\mathcal{I}(\mathbb{P})$. We discuss this in more detail in Appendix E.

Table 1: **Results on Erdos-Renyì (ER) graphs.** *First block:* We sample ten different ER random graphs, and accompanying non-linear structural equations as in Zheng et al. [36]. From each system we then sample a varying number of samples, and evaluate NOTEARS-MLP *with* D-Struct (indicated as "✓") and *without* D-Struct (indicated as "✗"). *Second block:* For each row we sample a new ER graph with a varying degree of connectedness ($s$ indicates the expected number of edges). In both cases, we report the average performance in terms of SHD, FPR, TPR, and FDR, with std in scriptsize. Unless otherwise indicated, $n = 1000, d = 5, K = 3, s = 2d$.

| *metric* | **SHD** ($\downarrow$) | | **FPR** ($\downarrow$) | | **TPR** ($\uparrow$) | | **FDR** ($\downarrow$) | |
|---|---|---|---|---|---|---|---|---|
| *D-Struct* | ✓ | ✗ | ✓ | ✗ | ✓ | ✗ | ✓ | ✗ |
| $n$ | | | | *varying sample size* | | | | |
| 200 | **3.60**±0.27 | 4.20±0.44 | **2.00**±0.67 | 4.20±0.44 | **0.67**±0.05 | 0.64±0.05 | **0.25**±0.06 | 0.42±0.04 |
| 500 | **3.20**±0.80 | 3.94±0.33 | **1.20**±0.44 | 3.94±0.33 | **0.66**±0.24 | 0.56±0.04 | **0.18**±0.05 | 0.44±0.04 |
| 1000 | **2.75**±0.47 | 3.67±0.82 | **1.00**±0.37 | 2.67±0.63 | **0.75**±0.08 | 0.63±0.13 | **0.18**±0.03 | 0.39±0.11 |
| 2000 | **2.66**±0.80 | 3.54±0.16 | **1.88**±0.67 | 2.09±0.31 | **0.81**±0.11 | 0.75±0.03 | **0.27**±0.07 | 0.33±0.00 |
| $s$ | | | | *varying graph connectedness* | | | | |
| 0.5$d$ | **3.75**±1.6 | 7.33±0.13 | **0.50**±0.25 | 1.05±0.02 | 0.83±0.19 | **0.88**±0.04 | 0.42±0.16 | 0.73±0.01 |
| 1$d$ | **3.50**±0.86 | 7.67±0.45 | **0.55**±0.22 | 1.53±0.09 | **0.75**±0.09 | 0.46±0.09 | **0.40**±0.09 | 0.77±0.07 |
| 1.5$d$ | **3.00**±1.15 | 5.67±1.75 | **1.00**±0.19 | 1.55±0.08 | **0.89**±0.07 | 0.62±0.06 | **0.32**±0.05 | 0.53±0.04 |
| 2$d$ | **2.28**±0.80 | 3.67±0.82 | **1.00**±0.32 | 2.67±0.63 | 0.67±0.17 | **0.70**±0.09 | **0.11**±0.03 | 0.32±0.08 |
| $d$ | | | | *varying dimension count* | | | | |
| 5 | **2.28**±0.80 | 3.67±0.82 | **1.00**±0.32 | 2.67±0.63 | 0.67±0.17 | **0.70**±0.09 | **0.11**±0.03 | 0.32±0.08 |
| 7 | **8.67**±0.56 | 12.9±0.15 | **0.72**±0.05 | 1.07±0.01 | **0.96**±0.02 | 0.83±0.01 | **0.49**±0.01 | 0.63±0.01 |
| 10 | **19.71**±0.72 | 30.8±0.98 | **0.42**±0.13 | 1.18±0.04 | 0.70±0.16 | **0.71**±0.06 | **0.34**±0.08 | 0.70±0.02 |

# 4 EXPERIMENTS

Recall from Section 3 that D-Struct's objective is exactly the same as any differentiable structure learner: transform a dataset into a DAG, whilst remaining differentiable. With D-Struct, our aim is to increase performance of any DSF by enforcing transportability on the learner's outcome structure. As such, the most pressing questions are: *(1) Are the discovered structures transportable?*, *(2) Does D-Struct improve existing learners?*, *(3) Does D-Struct give up on efficiency?*, and *(4) Do we really need our subsampling routine?* We answer these questions one-by-one below with experiments.

**(1) Transportability.** Before testing accuracy, we first empirically confirm that NOTEARS is not transportable while D-Struct is. We compare NOTEARS with D-Struct using 1000 samples drawn from an Erdos-Renyì (ER) random graph, and split the samples into two equal sized subsets. We evaluate the structural Hamming distance (SHD) between the graphs learned by NOTEARS on each dataset, and the same for the internal graphs learned by D-Struct. The DAGs learnt by D-Struct are perfectly transportable (SHD= 0) in 8/10 runs (mean SHD $0.46 \pm 0.27$), with only minor discrepancies in the other cases. Conversely, NOTEARS has a mean SHD of $1.14 \pm 0.20$, only displaying transportability in 2 cases. Similar results for other DSFs are reported in Appendix A.

**(2) Accuracy.** The most straightforward way to see if D-Struct is better is by repeating the experiments in Zheng et al. [36]. We report only a subset of our outcomes in the main text, mainly on D-Struct's improvement over NOTEARS-MLP. However, more metrics and experiments on different DSFs can be found in Appendix A. In Table 1 we report the false positive rate (FPR), true positive rate (TPR), false discovery rate (FDR), and structural Hamming distance (SHD) of the estimated DAGs using data sampled from different ER random graphs with varying sample size ($n$), expected number of edges ($s$), and dimension count ($d$). In all cases we find that D-Struct significantly improves NOTEARS-MLP (other DSFs in Appendix A). A similar conclusion can be drawn from Fig. 6, where we report the SHD for more parameters and data from Erdos-Renyì as well as Scale Free graphs [35].

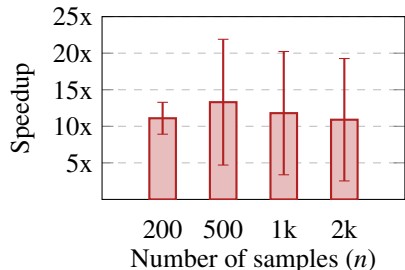

Figure 5: **Speedup of D-Struct over NOTEARS.** Difference in computation time between NOTEARS-MLP and D-Struct, as a function of $n$. On average, D-Struct is about 10x quicker.

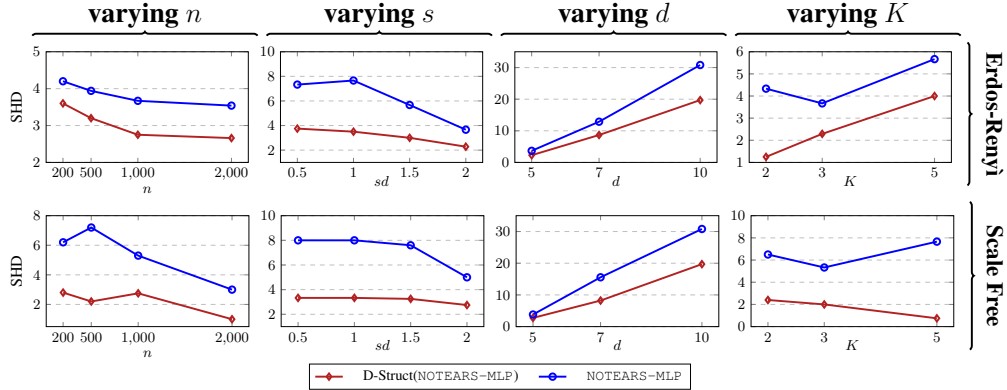

Figure 6: **Structure recovery.** We report the SHD (↓) compared to the true graph. We report performance as a function of four different parameters (changing the properties of the task). The results demonstrate that D-Struct outperforms NOTEARS-MLP in all these settings. Additional results are reported in Appendix A. Unless otherwise indicated, $n = 1000, d = 5, K = 3, s = 2d$.

---

**(3) Computational efficiency.** In Fig. 5 we learn that despite its parallel ensemble architecture, D-Struct is actually *much* faster than NOTEARS. Note that D-Struct is built *on top* of NOTEARS, meaning this computational gain is not due to differences in implementation. Instead, we believe computation gains are largely due to D-Struct's learning scheme. Rather than using the *entire* dataset at once to learn one (computationally intensive) DSF, D-Struct splits the data and learns multiple DSFs from several *smaller* datasets. We believe this is an important result: the whole reason for having differentiable structure learners is due to their efficiency gains over CIT-based methods.

**(4) Subset construction.** A final property we wish to validate is the need for sampling $K$ different subsets using our subsampling routine from Section 3.2. This is an important validation as it shows that D-Struct does not *only* gain in performance due to its ensemble architecture. For this, we compare D-Struct's performance *with* and *without* our subsampling routine. Using D-Struct without our subsampling routine amounts to providing $K$ random splits, rather than carefully sampling $K$ distinct $\mathcal{D}_k \sim \mathbb{P}^k$. Table 2 shows that our subsampling routine *does* improve D-Struct's performance as expected, validating our goal to explicitly optimise for transportable structure learners.

We believe that these experiments offer a broad view of how D-Struct and transportability can help us create useful structure learners. However, as we were unable to include all our experimental results in the main text of this paper, we refer the interested reader to our Appendix A for completed and additional validation. In our appendices we also include an anonymous link to our code repository, encouraging readers to reproduce our results.

Table 2: **Usefulness of our subsampling routine.** We sample ten different ER random graphs like in Table 1. From each system we then sample $n = 2000$ samples, and evaluate NOTEARS-MLP *with* our subsampling routine from Section 3.2 (indicated as "✓") and *without* the subsampling routine, using random splits instead (indicated as "✗"). For each row we repeat our experiment with different $K$. We report the average (and std) performance in terms of the SHD.

| metric | SHD (↓) | |
|---|---|---|
| *Subsample* | ✓ | ✗ |
| $K$ | *varying amount of splits* | |
| 2 | **2.80**±0.53 | 3.40±0.58 |
| 3 | **3.00**±0.37 | 4.00±0.59 |
| 5 | **2.80**±0.57 | 4.40±1.29 |

## 5 DISCUSSION

D-Struct advances differentiable structure learning by introducing transportability, a property guaranteed by CIT-based methods. We believe D-Struct can have a positive impact on architectures and problems relying on differentiable structure learners, as well as on general scientific data analysis.

**Relating DSFs to causality.** As pointed out by Kaiser and Sipos [69] and Reisach et al. [70], DSFs are often wrongly used to recover a *causal* DAG. While DAGs are indeed the model of choice to describe causal interactions, there is currently no guarantee that a DAG discovered using any DSF can be identified (and thus used) as such. With this we wish to state explicitly that a DSF's output is *not* to be interpreted as a causal model. We discuss more about this particular topic in Appendix B.

**Future work.** The inability to recover causal structure is a consequence of there existing many more useful properties stemming from a CIT-based approach (multiple books concern this very topic [26, 53, 71, 72]). Bridging the gap between these methods is a clear path forward, hopefully

increasing differentiable structure learners' potential even further. Specifically, using structure learners to uncover a causal structure from observational data requires stricter assumptions. As such, one particularly interesting avenue of future work is to allow DSFs (not only D-Struct) to adhere to some of these assumptions and use them to guarantee causal discovery, taking DSFs to a next level.

Finally, D-Struct is only the *first step* in the process of scientific discovery. As D-Struct (or any other DSF) *suggests* a link between variables, it is still the scientist's task to confirm this link in the lab.

**Ethics Statement.** We envisage D-Struct as a tool to *help* the scientific endeavour, however emphasise the discovered structures and links would need to be further verified by a human expert or in an experimental setting. Furthermore, the data used in this work is synthetically generated from given random graphical models, and no human-derived data was used.

**Reproducibility Statement.** To ensure reproducibility, we include experimental details in Appendix A.1. These details include: (1) hyperparameter settings, (2) evaluation metrics, (3) the synthetic data generation procedure, and (4) additional implementation details of D-Struct. Finally, all code is readily available at our anonymous online code repository: `https://anonymous.4open.science/r/d-struct`. Beyond documentation and instructions, this code includes benchmark models, synthetic data generation, and our D-Struct implementation.

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

# Appendix: D-Struct

## Table of Contents

## A    ADDITIONAL EXPERIMENTS

Please find our (anonymous) online code repository at:

Our code is based on code provided by Zheng et al. [36], and we annotated our code where we used their implementation.

### A.1    SETTINGS AND DETAILS

In the interest of space, we left out a few details in our main text. Here we discuss hyperparameters (those in addition to the hyperparameters required for the selected DSFs), the evaluation metrics, and how we combine the different parallel DAGs.

**Hyperparameters.**  D-Struct inherits hyperparameters from the chosen underlying DSFs. These hyperparameters act in the same way as they would in their original incarnation. For a discussion on these hyperparameters we refer to the relevant literature on these methods specifically.

However, D-Struct also adds two additional parameters: $K$ and $\alpha$. The impact of $K$ is already discussed in the main text, recapitulated as: $K$ implicitly determines the sizes of the subsets used to train the parallel DSFs, as such, *for high $K$ we should have high $n$*. With both increasing, we report better performance (particularly in Scale-Free DAGs).

The impact of $\alpha$ is a bit more subtle, and also a function of $K$. First, consider Fig. 7, displaying the impact on each evaluation metric as a function of different $\alpha$. What we find is that setting $\alpha$ is mostly

dependent on $K$ as lower $\alpha$ tend to work better with higher $K$, and vice versa for lower $K$. This makes sense as we sum each $\mathcal{L}_{\mathbf{MSE}}$, resulting in a higher value with more $K$. If $\alpha$ is large in a setting with large $K$, the regularisation effect would simply be too large. We set our hyperparameters to those which yielded best performance (deduced from Fig. 7 for $\alpha$, and $K = 3$ when not varied over as this yielded most stable results overall).

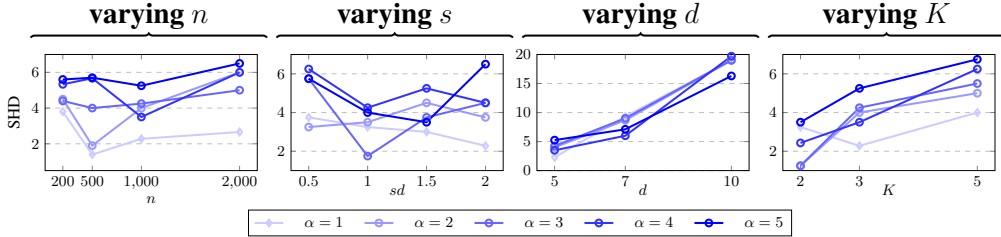

Figure 7: **Results showing the effect of** $\alpha$. Depending on the nature of the problem the degree of regularization imposed by $\alpha$ can vary. This then changes the amount we enforce the similarity between the different D-Struct adjacency. matrices.

**Evaluation metrics.** The learned graphs from NOTEARS and D-Struct are assessed using four graph metrics namely: (1) Structural Hamming distance (SHD), (2) False discovery rate (FDR), (3) False positive rate (FPR) and (4) True positive rate (TPR). These values are standard when evaluating structure learning methods. We provide some insight into these evaluation metrics below.

*Structural Hamming distance (SHD)*     SHD is the total number of edge additions, deletions, and reversals needed to convert the estimated DAG into the true DAG. That means that the worst case SHD is $d^2 - d$, as we bound the diagonal to be 0 at all times. As such, the reported SHD with varying $d$ is expected to be higher, not due to hardness of the problem, but as a property of the SHD (see for example Fig. 6).

*False discovery rate (FDR)*     Whenever an edge is suggested in the estimated DAG, which is incorrect, we add to the falsely discovered edges. As such, the FDR is defined as the number of reversed edges and edges that should not exist, divided by the number of edges in total. Of course, the exception being when no edges are suggested at all (which implies dividing by 0), which naturally has an FDR of zero.

*False positive rate (FPR)*     We sum the edges that should have been reversed and those that should not exist, and divide by the total number of *non-edges* in the ground truth DAG. A non-edge is an edge that does not exist. With a more connected ground truth DAG, we expect this number to be lower automatically (as the numerator of the FPR would be higher). This is the reason why we let $s$ be a function of $d$, as increasing the number of expected edges with $d$ would somewhat counter this effect. Note that, in Table 1 we see the FPR increasing proportionate to the factor multiplied with $d$, which is as we would expect.

*True positive rate (TPR)*     This signifies the number of correctly estimated edges, over the number of edges in the true graph. Note that, reversed edges are counted as wrong edges.

**Combining graphs.** Inference is done by combining the $K$ internal graphs. In our implementation of D-Struct we combine graphs by averaging the adjacency matrices and apply a threshold to convert the average graph into a binary matrix. The latter is a similar strategy to most DSFs' strategies to convert a continuous matrix into a binary one. This is a relatively simple method with promising results, in line with what is currently done in literature.

However, given that D-Struct has multiple graphs, we can actually come up with different strategies (a potential topic for future research). Naturally, this would be more relevant with high $K$, which in turn requires a larger sample-size, as per our discussion above. Specifically, we enter the domain of ensemble learning. Like D-Struct, ensemble methods need to combine, potentially conflicting, outcomes and provide the user with only one outcome.

Table 3: **Results on Erdos-Renyì (ER) graphs.** *First block:* We sample five different ER random graphs, and accompanying non-linear structural equations using an index-model. From each system we then sample a varying number of samples, and evaluate `NOTEARS-MLP` *with* D-Struct (indicated as "✓") and *without* D-Struct (indicated as "✗"). *Second block:* For each row we sample a new ER graph with a varying degree of connectedness ($s$ indicates the expected number of edges). In both cases, we report the average performance in terms of SHD, FPR, TPR, and FDR, with std in scriptsize.

| metric | SHD ($\downarrow$) | | FPR ($\downarrow$) | | TPR ($\uparrow$) | | FDR ($\downarrow$) | |
|---|---|---|---|---|---|---|---|---|
| D-Struct | ✓ | ✗ | ✓ | ✗ | ✓ | ✗ | ✓ | ✗ |
| $n$ | *varying sample size* | | | | | | | |
| 200 | **3.60**±0.27 | 4.20±0.44 | **2.00**±0.67 | 4.20±0.44 | **0.67**±0.05 | 0.64±0.05 | **0.25**±0.06 | 0.42±0.04 |
| 500 | **3.20**±0.80 | 3.94±0.33 | **1.20**±0.44 | 3.94±0.33 | **0.66**±0.24 | 0.56±0.04 | **0.18**±0.05 | 0.44±0.04 |
| 1000 | **2.75**±0.47 | 3.67±0.82 | **1.00**±0.37 | 2.67±0.63 | **0.75**±0.08 | 0.63±0.13 | **0.18**±0.03 | 0.39±0.11 |
| $s$ | *varying graph connectedness* | | | | | | | |
| $0.5d$ | **3.75**±1.6 | 7.33±0.13 | **0.50**±0.25 | 1.05±0.02 | 0.83±0.19 | **0.88**±0.04 | **0.42**±0.16 | 0.73±0.01 |
| $1d$ | **3.50**±0.86 | 7.67±0.45 | **0.55**±0.22 | 1.53±0.09 | **0.75**±0.09 | 0.46±0.09 | **0.40**±0.09 | 0.77±0.07 |
| $1.5d$ | **3.00**±1.15 | 5.67±1.75 | **1.00**±0.19 | 1.55±0.08 | **0.89**±0.07 | 0.62±0.06 | **0.32**±0.05 | 0.53±0.04 |
| $2d$ | **2.28**±0.80 | 3.67±0.82 | **1.00**±0.32 | 2.67±0.63 | 0.67±0.17 | **0.70**±0.09 | **0.11**±0.03 | 0.32±0.08 |
| $d$ | *varying dimension count* | | | | | | | |
| 5 | **2.28**±0.80 | 3.67±0.82 | **1.00**±0.32 | 2.67±0.63 | 0.67±0.17 | **0.70**±0.09 | **0.11**±0.03 | 0.32±0.08 |
| 7 | **8.67**±0.56 | 12.88±0.15 | **0.72**±0.05 | 1.07±0.01 | **0.96**±0.02 | 0.83±0.01 | **0.49**±0.01 | 0.63±0.01 |
| 10 | **19.71**±0.72 | 30.82±0.98 | **0.42**±0.13 | 1.18±0.04 | 0.70±0.16 | **0.71**±0.06 | **0.34**±0.08 | 0.70±0.02 |

One avenue is to not vote on a per-element basis, but on a per-graph basis. Imagine, two graphs in $K$ that are exactly the same aspire more confidence in their accuracy. We could even relax similarity to an SHD across graphs, where we weight each graph's "vote" proportionally to their combined SHD. We believe this to be promising area of future research.

**Experimental procedure.** Here we explain how our experimental setup works, which steps we need to perform before starting an experiment, and which information each model is provided.

There are two main parts to an experimental setup: (i) we need a structure, (ii) we need a set of structural equations accompanying the structure of step (i).

*(i) The structure.* In our setup, a structure can only be a DAG. To reduce bias as much as possible, we do not determine structures up front, but sample random structures for each experimental run. Of course, the same random structure is presented for each benchmark. Sampling random structures happens in two ways: either we sample a random Erdös-Renyi graph, which requires a dimension count ($d$), and an expected number of edges ($ds$); or we use a scale-free graph which is generated using the process described in Barabási and Albert [73] as was also done in Zheng et al. [36], which needs a parameter $\beta = 1$ (the exponent for the preferential attachment process). The expected number of edges in our setup depends on $d$ such that $s$ resembles the ratio of edges versus non-edges in the random graph.

*(ii) The equations.* With a sampled structure from (i), we can now sample some structural equations. In our paper we use an index model to sample these. In short, an index model is randomly parameterised as: $f_j(X_{\text{pa}(j)}) = \sum_{m=1}^3 h_m(\sum_{k \in \text{pa}(j)} \theta_{jmk} X_k)$, where $h_1 = \tanh$, $h_2 = \cos$, $h_3 = \sin$, and each $\theta_{jmk}$ is drawn uniformly from range $[-2, -0.5] \cup [0.5, 2]$. Exactly as was reported in Zheng et al. [36].

## A.2 COMPLETED RESULTS

Recall from Section 4 that we only reported a subset of the results. In Table 5 we report the remainder for `NOTEARS-MLP` and the D-Struct implementation on scale free graphs.

Table 4: **Results on Scale-Free (SF) graphs.** *First block:* We sample five different SF random graphs, and accompanying non-linear structural equations using an index-model. From each system we then sample a varying number of samples, and evaluate NOTEARS-SOB *with* D-Struct (indicated as "✓") and *without* D-Struct (indicated as "✗"). *Second block:* For each row we sample a new SF graph with a varying degree of connectedness ($s$ indicates the expected number of edges). *Third block:* For each row we vary the feature dimension count ($d$). *Fourth block:* For each row we vary the number of subsets for D-Struct ($s$).In all cases, we report the average performance in terms of SHD, FPR, TPR, and FDR, with std in scriptsize.

| *metric* | **SHD** ($\downarrow$) | | **FPR** ($\downarrow$) | | **TPR** ($\uparrow$) | | **FDR** ($\downarrow$) | |
|---|---|---|---|---|---|---|---|---|
| *D-Struct* | ✓ | ✗ | ✓ | ✗ | ✓ | ✗ | ✓ | ✗ |
| $n$ | | | | *varying sample size* | | | | |
| 200 | $6.00 \pm 0.69$ | $3.8 \pm 0.25$ | $1.8 \pm 0.20$ | $1.27 \pm 0.08$ | $0.49 \pm 0.12$ | $0.83 \pm 0.03$ | $0.63 \pm 0.08$ | $0.39 \pm 0.02$ |
| 500 | $\mathbf{3.40} \pm 0.88$ | $4.60 \pm 0.25$ | $\mathbf{0.67} \pm 0.14$ | $1.53 \pm 0.08$ | $0.57 \pm 0.09$ | $0.69 \pm 0.03$ | $\mathbf{0.41} \pm 0.12$ | $0.48 \pm 0.03$ |
| 1000 | $\mathbf{2.75} \pm 0.86$ | $4.33 \pm 0.50$ | $\mathbf{0.58} \pm 0.22$ | $1.44 \pm 0.17$ | $0.61 \pm 0.15$ | $0.76 \pm 0.05$ | $\mathbf{0.36} \pm 0.15$ | $0.44 \pm 0.05$ |
| $s$ | | | | *varying graph connectedness* | | | | |
| $0.5d$ | $\mathbf{14.11} \pm 5.40$ | $39.53 \pm 0.37$ | $\mathbf{0.31} \pm 0.13$ | $0.89 \pm 0.01$ | $\mathbf{0.22} \pm 0.14$ | $0.20 \pm 0.11$ | $\mathbf{0.42} \pm 0.17$ | $0.93 \pm 0.01$ |
| $1d$ | $\mathbf{8.11} \pm 3.96$ | $39.46 \pm 0.38$ | $\mathbf{0.13} \pm 0.09$ | $0.89 \pm 0.01$ | $\mathbf{0.30} \pm 0.18$ | $0.16 \pm 0.09$ | $\mathbf{0.22} \pm 0.15$ | $0.99 \pm 0.01$ |
| $1.5d$ | $\mathbf{15.20} \pm 3.44$ | $38.31 \pm 0.41$ | $\mathbf{0.32} \pm 0.14$ | $1.05 \pm 0.01$ | $\mathbf{0.58} \pm 0.23$ | $0.52 \pm 0.07$ | $\mathbf{0.40} \pm 0.17$ | $0.98 \pm 0.01$ |
| $2d$ | $\mathbf{15.20} \pm 3.44$ | $38.25 \pm 0.44$ | $\mathbf{0.32} \pm 0.14$ | $1.04 \pm 0.01$ | $\mathbf{0.58} \pm 0.23$ | $0.50 \pm 0.07$ | $\mathbf{0.40} \pm 0.17$ | $0.89 \pm 0.01$ |
| $d$ | | | | *varying dimension count* | | | | |
| 5 | $\mathbf{2.75} \pm 0.86$ | $4.33 \pm 0.50$ | $\mathbf{0.58} \pm 0.22$ | $1.44 \pm 0.17$ | $0.61 \pm 0.15$ | $0.76 \pm 0.05$ | $\mathbf{0.36} \pm 0.15$ | $0.44 \pm 0.05$ |
| 7 | $\mathbf{8.25} \pm 3.09$ | $15.00 \pm 0.22$ | $\mathbf{0.55} \pm 0.21$ | $1.00 \pm 0.01$ | $\mathbf{0.96} \pm 0.08$ | $0.78 \pm 0.02$ | $\mathbf{0.49} \pm 0.16$ | $0.76 \pm 0.01$ |
| 10 | $\mathbf{16.80} \pm 4.21$ | $35.75 \pm 0.33$ | $\mathbf{0.36} \pm 0.17$ | $0.99 \pm 0.01$ | $0.58 \pm 0.24$ | $0.67 \pm 0.03$ | $\mathbf{0.42} \pm 0.17$ | $0.85 \pm 0.01$ |
| $K$ | | | | *varying subset count* | | | | |
| 2 | $\mathbf{3.00} \pm 0.42$ | $6.00 \pm 0.30$ | $\mathbf{0.53} \pm 0.21$ | $2.00 \pm 0.10$ | $\mathbf{0.66} \pm 0.04$ | $0.57 \pm 0.04$ | $\mathbf{0.21} \pm 0.07$ | $0.60 \pm 0.03$ |
| 3 | $\mathbf{2.75} \pm 0.86$ | $4.33 \pm 0.5$ | $\mathbf{0.58} \pm 0.22$ | $1.44 \pm 0.17$ | $0.61 \pm 0.15$ | $0.76 \pm 0.05$ | $\mathbf{0.36} \pm 0.15$ | $0.44 \pm 0.05$ |
| 5 | $\mathbf{2.80} \pm 0.57$ | $5.25 \pm 0.21$ | $\mathbf{0.73} \pm 0.15$ | $1.75 \pm 0.07$ | $\mathbf{0.74} \pm 0.09$ | $0.68 \pm 0.03$ | $\mathbf{0.31} \pm 0.07$ | $0.53 \pm 0.02$ |

## A.3    OTHER DSFS

We repeat the results above for NOTEARS-SOB which is a Sobolev based implementation of NOTEARS, in Table 4. The main difference here with NOTEARS-MLP is the nonparametric estimation of the structural equations in $\hat{\mathcal{G}}$. Note that, future implementations of DSFs broadly alter the way in which the structural equations are estimated, and much less on how the proposed structure is evaluated to be a DAG (as they are mostly based on eq. (3)). Overall, we find that NOTEARS-SOB behaves the same as NOTEARS-MLP: D-Struct vastly improves performance.

Note that code to reproduce above results is provided in the online code repository linked to above.

## A.4    SUBSAMPLING DATASETS

We refer to Table 6 for the full results presented originally in Table 2. While FPR may be a little higher, using D-Struct still outperforms not using D-Struct in terms of the FPR– already shown in Table 1. Furthermore, as the subsampling routine forces D-Struct to learn on different distributions, it is possible that this increase in FPR is a result of initially more conflicting DAG structures. When combined, these structures include more edges which in turn result in more potential for a false positive edge discovery. In fact, we observe a lower necessary threshold when using our subsampling routine, necessary to transform the real-values matrix into a binary adjacency matrix.

We also report the same metrics as a function of the DAG-finding threshold in Fig. 8, where the threshold is applied to the adjacency matrix to produce a binary matrix on which we compute the metrics. Of course, a threshold will be selected in practice; however, we show that for a range of plausible threshold values and all metrics that subsampling with our routine is indeed beneficial, compared to randomized subsampling. From this, it seems that the results we find in Table 6 are consistent even with changing thresholds.

Table 5: **Results on Scale-Free (SF) graphs.** *First block:* We sample five different SF random graphs, and accompanying non-linear structural equations using an index-model. From each system we then sample a varying number of samples, and evaluate `NOTEARS-MLP` *with* D-Struct (indicated as "✓") and *without* D-Struct (indicated as "✗"). *Second block:* For each row we sample a new SF graph with a varying degree of connectedness ($s$ indicates the expected number of edges). *Third block:* For each row we vary the feature dimension count ($d$). *Fourth block:* For each row we vary the number of subsets for D-Struct ($s$). In all cases, we report the average performance in terms of SHD, FPR, TPR, and FDR, with std in scriptsize.

| metric | SHD ($\downarrow$) | | FPR ($\downarrow$) | | TPR ($\uparrow$) | | FDR ($\downarrow$) | |
|---|---|---|---|---|---|---|---|---|
| D-Struct | ✓ | ✗ | ✓ | ✗ | ✓ | ✗ | ✓ | ✗ |
| $n$ | *varying sample size* | | | | | | | |
| 200 | **2.80**±0.86 | 6.20±0.57 | **0.73**±0.28 | 2.07±0.19 | **0.80**±0.11 | 0.54±0.08 | **0.26**±0.11 | 0.62±0.06 |
| 500 | **2.20**±0.80 | 7.20±0.66 | **0.27**±0.12 | 2.20±0.18 | **0.77**±0.13 | 0.37±0.09 | **0.14**±0.06 | 0.72±0.06 |
| 1000 | **3.25**±1.49 | 5.33±0.61 | **0.75**±0.43 | 1.78±0.20 | **0.68**±0.15 | 0.66±0.08 | **0.29**±0.18 | 0.53±0.06 |
| $s$ | *varying graph connectedness* | | | | | | | |
| $0.5d$ | **3.33**±0.88 | 8.00±0.37 | **0.50**±0.19 | 1.17±0.06 | **0.92**±0.08 | 0.38±0.05 | **0.41**±0.08 | 0.82±0.03 |
| $1d$ | **3.33**±0.89 | 8.00±1.00 | **0.50**±0.19 | 1.17±0.17 | **0.92**±0.08 | 0.38±0.13 | **0.41**±0.08 | 0.82±0.07 |
| $1.5d$ | **3.25**±0.41 | 7.67±0.31 | **0.50**±0.07 | 1.17±0.04 | **0.94**±0.04 | 0.42±0.06 | **0.43**±0.03 | 0.80±0.03 |
| $2d$ | **2.75**±1.03 | 5.00±1.00 | **0.33**±0.23 | 1.22±0.22 | **0.64**±0.15 | 0.50±0.07 | **0.14**±0.09 | 0.48±0.12 |
| $d$ | *varying dimension count* | | | | | | | |
| 5 | **3.25**±1.49 | 5.33±0.61 | **0.75**±0.43 | 1.78±0.20 | **0.68**±0.15 | 0.66±0.08 | **0.29**±0.18 | 0.53±0.06 |
| 7 | **8.22**±1.31 | 15.67±0.14 | **0.54**±0.09 | 1.04±0.01 | **0.98**±0.02 | 0.83±0.03 | **0.54**±0.04 | 0.76±0.01 |
| 10 | **16.80**±4.21 | 35.75±0.33 | **0.36**±0.17 | 0.99±0.01 | 0.58±0.24 | 0.67±0.03 | **0.42**±0.17 | 0.85±0.01 |
| $K$ | *varying subset count* | | | | | | | |
| 2 | **2.40**±0.24 | 6.50±0.46 | **0.53**±0.08 | 2.16±0.15 | **0.83**±0.05 | 0.50±0.06 | **0.21**±0.03 | 0.65±0.06 |
| 3 | **2.00**±1.04 | 5.33±0.6 | **0.33**±0.47 | 1.78±0.20 | **0.68**±0.14 | 0.66±0.09 | **0.14**±0.09 | 0.53±0.06 |
| 5 | **0.75**±0.48 | 5.25±0.21 | **0.25**±0.16 | 2.55±0.11 | **1.00**±0.00 | 0.33±0.05 | **0.09**±0.05 | 0.76±0.03 |

Figure 8: **Subsampling with different DAG-thresholds.** The DAG-threshold transforms the real-valued adjacency matrix, to a binary one. As the threshold increases, the amount edges that remain part of the DAG decreases. The above confirms our findings from Table 6 in different settings.

## A.5 DAGS: D-STRUCT VS NOTEARS

We wish to also highlight that indeed what is recovered by D-Struct is different from NOTEARS. For this we refer to Figs. 9 and 10, each representing an independent run.

## A.6 GAINS FROM ENFORCING TRANSPORTABILITY

A key concept of D-Struct is to enforce transportability, which is done using our novel loss function.

$$\mathcal{L}(\mathcal{G}_k|\mathcal{D}_k) \coloneqq \mathcal{L}_{\text{DSF}}(\mathcal{G}|\mathcal{D}_k) + \alpha \mathcal{L}_{\text{MSE}}(A(\mathcal{G}_k)),$$

Table 6: **Usefulness of our subsampling routine.** We sample ten different ER random graphs, and accompanying non-linear structural equations as in Zheng et al. [36]. From each system we then sample $n = 2000$ samples, and evaluate NOTEARS-MLP *with* our subsampling routine from Section 3.2 (indicated as "✓") and *without* the subsampling routine, using random splits instead (indicated as "✗"). For each row we repeat our experiment with different $K$. In both cases, we report the average performance in terms of SHD, FPR, TPR, and FDR, with std in scriptsize.

| metric | **SHD** ($\downarrow$) | | **FPR** ($\downarrow$) | | **TPR** ($\uparrow$) | | **FDR** ($\downarrow$) | |
|---|---|---|---|---|---|---|---|---|
| Subsample | ✓ | ✗ | ✓ | ✗ | ✓ | ✗ | ✓ | ✗ |
| $K$ | | | | *varying amount of splits* | | | | |
| 2 | **2.80**±0.53 | 3.40±0.58 | 2.80±0.53 | **2.60**±0.33 | **0.80**±0.06 | 0.71±0.07 | **0.28**±0.05 | 0.30±0.16 |
| 3 | **3.00**±0.37 | 4.00±0.59 | 2.00±0.51 | **1.60**±0.45 | **0.73**±0.04 | 0.58±0.06 | **0.22**±0.05 | 0.24±0.17 |
| 5 | **2.80**±0.57 | 4.40±1.29 | 1.40±0.50 | **0.60**±0.26 | **0.71**±0.06 | 0.53±0.15 | 0.18±0.06 | **0.07**±0.10 |

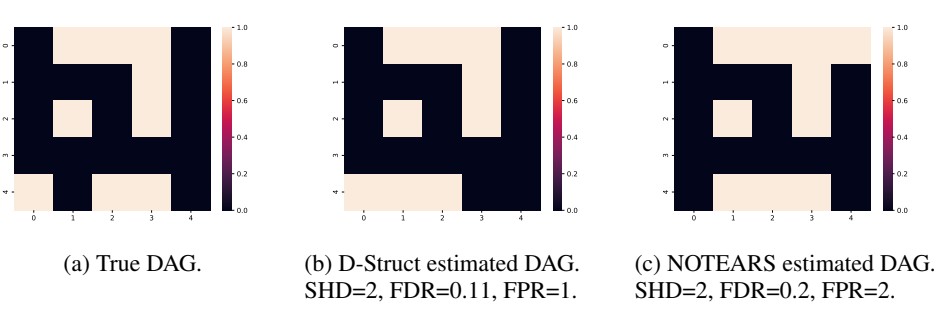

(a) True DAG.  |  (b) D-Struct estimated DAG. SHD=2, FDR=0.11, FPR=1.  |  (c) NOTEARS estimated DAG. SHD=2, FDR=0.2, FPR=2.

Figure 9: First independent run

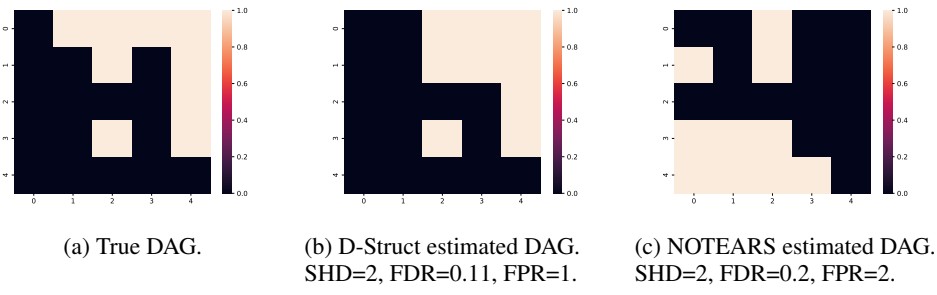

(a) True DAG.  |  (b) D-Struct estimated DAG. SHD=2, FDR=0.11, FPR=1.  |  (c) NOTEARS estimated DAG. SHD=2, FDR=0.2, FPR=2.

Figure 10: Second independent run

The question is what do we gain from the usage of the $\alpha$ term which is key to enforcing transportability. We conduct an experiment where we set $\alpha = 0$. This not only assesses the importance of this term, but also without $\mathcal{L}_{\text{MSE}}$ this amounts to assessing $K$ *independent* versions of vanilla NOTEARS.

*Results:* When we combine the $K$ DAGs by averaging them, the result is NOT a DAG.

This highlights that indeed that (1) transportability is key as part of this formulation and (2) that simply running parallel versions of NOTEARS is not a sufficient solution.

We highlight this by showing the independent DAGs discovered without transportability enforced, the average of the DAGs and the true DAG. These results are reported in Figs. 11 and 12

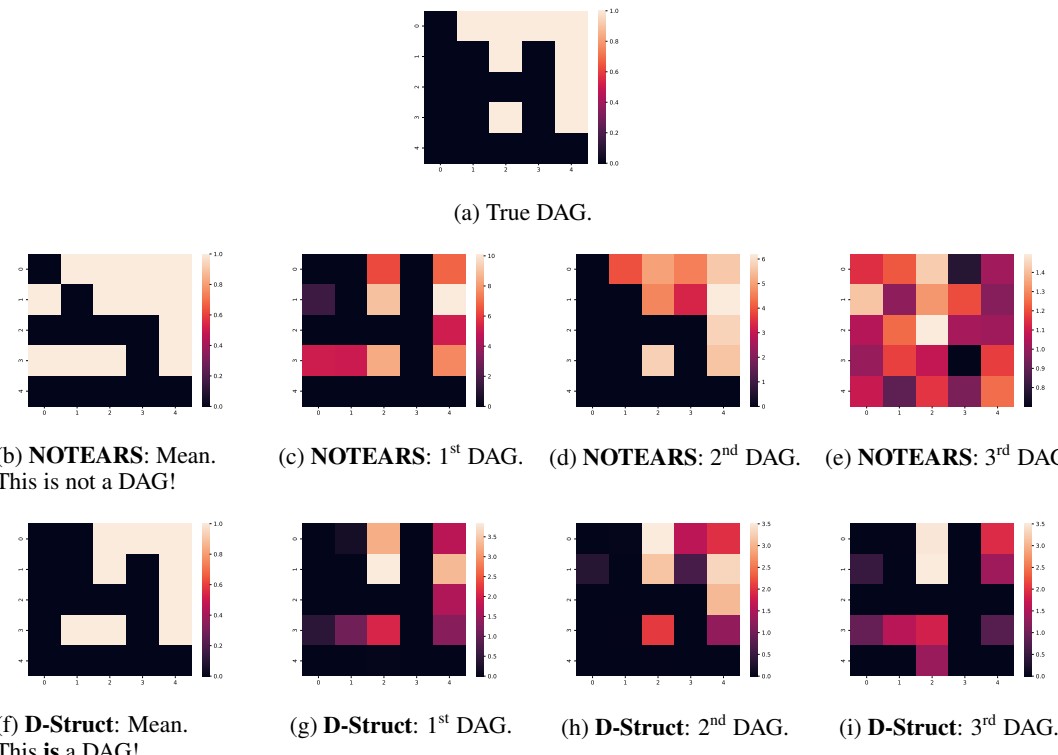

(a) True DAG.

(b) **NOTEARS**: Mean. This is not a DAG!

(c) **NOTEARS**: 1st DAG.

(d) **NOTEARS**: 2nd DAG.

(e) **NOTEARS**: 3rd DAG.

(f) **D-Struct**: Mean. This **is** a DAG!

(g) **D-Struct**: 1st DAG.

(h) **D-Struct**: 2nd DAG.

(i) **D-Struct**: 3rd DAG.

Figure 11: **First independent run.** Note the differences between the three DAGS on each partition for NOTEARS (Row 1), the average is also not a DAG. Whereas, for D-Struct note the similarities by enforcing transportability, the average is also a DAG.

## B  CAUSAL INTERPRETATION AND UNIQUENESS

**Causality.** Causal relationships between variables are often expressed as DAGs [28]. While D-Struct is able to recover DAGs more reliably, there is actually no guarantee that the found DAG can be interpreted as a causal DAG. There is a simple reason for this: we do not make any additional identification assumptions on the structural equations when learning DAGs, at least not beyond what is already assumed in the used DSFs. Furthermore, should D-Struct be combined with a DSF that *is* able to recover a causal DAG[4], the way in which the $K$ internal DAGs are combined may violate these assumptions (recall DAG combination from Appendix A.1).

With D-Struct, we recover a Bayesian network (BN), which is directed, yet the included directions are not necessarily meaningful. The only guarantee we have with BNs is that they resemble a distribution, which express some conditional distributions (as per the independence sets in Section 2.1). Order is not accounted for in these independence sets. For more information regarding this, we refer to Appendix D and Koller and Friedman [26].

However, as is indicated in Koller and Friedman [26, Chapter 21], a "good" BN structure should correspond to causality, where edges $X \to Y$ indicated that $X$ causes $Y$. Koller and Friedman [26] state that BNs with a causal structure tend to be sparser. Though, if queries remain probabilistic, it doesn't matter whether or not the structure is causal, the answers will remain the same. Only when we are interested in interventional queries (by using do-calculus) we have to make sure the DAG is a causal one.

**Uniqueness.** The above is a pragmatic view. To our knowledge, there is no real proof stating that sparser DAGs are (even more likely to be) causal. However, it could offer guidance to try and recover a causal DAG, assuming it to be sparse [76]. The latter of course is assuming that there exists a *unique* or *correct* DAG, which is something we implicitly assume to be true. Naturally, when

---

[4]We know of none that is able to.

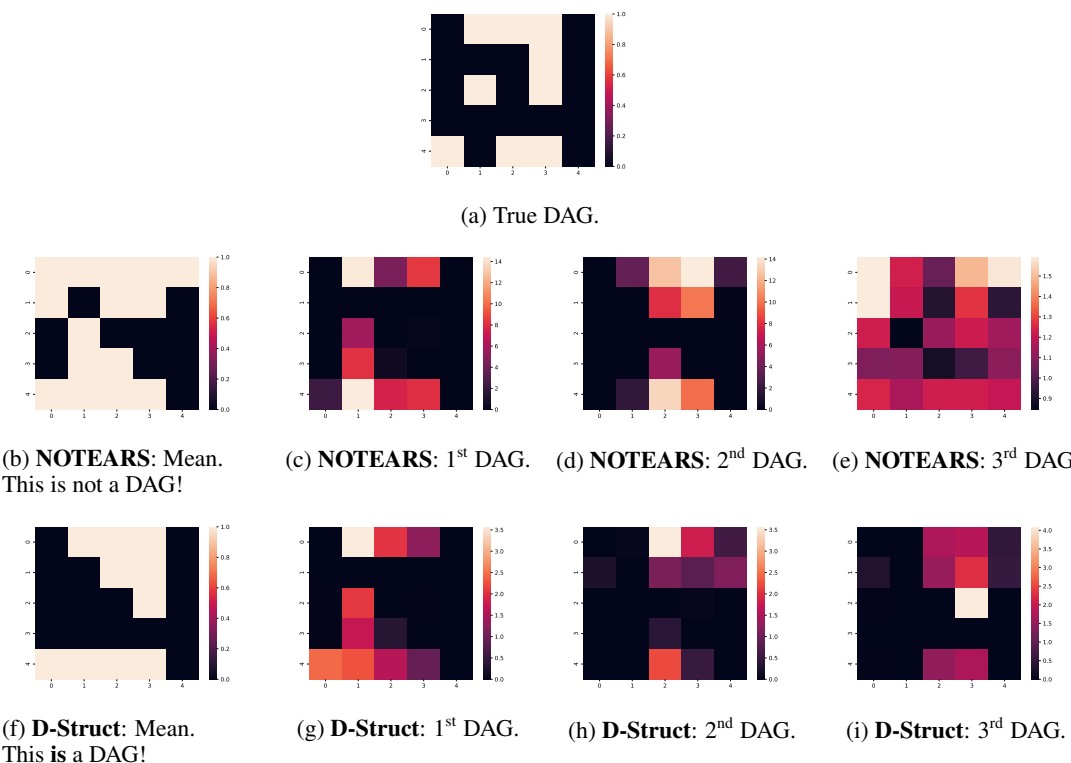

(a) True DAG.

(b) **NOTEARS**: Mean. This is not a DAG!

(c) **NOTEARS**: 1st DAG.

(d) **NOTEARS**: 2nd DAG.

(e) **NOTEARS**: 3rd DAG.

(f) **D-Struct**: Mean. This **is** a DAG!

(g) **D-Struct**: 1st DAG.

(h) **D-Struct**: 2nd DAG.

(i) **D-Struct**: 3rd DAG.

Figure 12: **Second independent run.** Note the differences between the three DAGS on each partition for NOTEARS (Row 1), the average is also not a DAG. Whereas, for D-Struct note the similarities by enforcing transportability, the average is also a DAG.

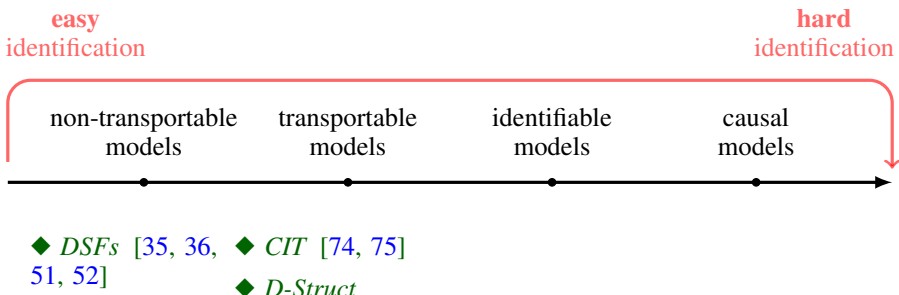

Figure 13: **Comparison of methods w.r.t. identification and uniqueness.** The ultimate goal of structure learning is to come up with unique and correct structures. Once we recover the one true DAG, we may interpret the structure as a causal model. However, discovering a causal structure using only observational data is not possible. Yet, we can *approach* it with methods that restrict the set of possible DAGs. From this illustration, we gather that D-Struct is an attempt to restrict the solution space of DSFs, going one step further towards unique solutions.

aiming to make a discovery, we aim to recover a *true* DAG, where a truthful DAG is corresponding with a DAG that can be uniquely recovered.

However, there is a difference between *a* unique DAG, and *the* unique DAG. Where the former is a matter of identifiability (discussed more below), the latter is one of causality. With the latter we mean: "can a method actually recover the unique causal DAG?" From Meek [39] and Meek [39] we learn that, from observational data alone, this is impossible and should thus not be a goal if one is not willing to make additional assumptions.

We stress that transportability is a weaker goal than identifiability. Enforcing transportability does not guarantee unique or repeatable results. Take CIT-based methods— which we know to be fully transportable. While it is true that the same set of independence statements will always result in the same DAG (i.e. transportability), it is not necessarily true that we will always recover the same independence statements. Depending on which independence test one uses to build the set of independence statements, the resulting DAG may look entirely different. Similar for D-Struct, while D-Struct does encourage similar DAGs (see for example Appendix A), we have no guarantee to recover the *same* DAG over different runs. The latter is a requirement for identifiability [77] as identifiability requires the model to always converge to the same set of parameters.

However, we do believe transportability is a vehicle to bring us closer to unique identification with DSFs. It is clear from our experiments that transportable learners greatly improve edge accuracy. As our synthetic setup is governed by one (and thus unique) graph, having a more accurate learner means a learner that discovers a DAG that is more like the unique, underlying graphical model. Consider Fig. 13 for an illustration comparing the relevant methods in terms of model identification.

## C    TRANSPORTABILITY IN NON-OVERLAPPING DOMAINS

Consider the multi-origin setting, where we have at least two datasets, each stemming from a different source. It is entirely possible that, given the different sources, these datasets are not comparable in terms of recorded features. We can recognise two major manifestations of this phenomenon: either (i) the supports of the datasets do not match, or (ii) the dimensions do not match.

**(i) Different support.** Recall from Section 2.1 that DAGs encode a set of independence statements. As such, it is mainly independence that governs structure. Transportability in the setting of conflicting support, thus requires some (mild) assumptions. Specifically, we require that independence holds, regardless of support. The is mostly a pragmatic assumption. If for example, we find that $\mathcal{X}_i \perp\!\!\!\perp \mathcal{X}_j$, where each component denotes a dimension in $\mathcal{X}$, we usually don't specify over what support this independence holds. Implicitly, we assume that independence holds, regardless of what area in $\{\mathcal{X}_i, \mathcal{X}_j\}$ we find ourselves in.

Note that the chosen distributions in $\mathcal{P}$ in Section 3.2 govern the entire domain $[N]$, and as a consequence $\mathcal{X}$. As such, the problem of conflicting support does not manifest in our solution of single-origin D-Struct. In case one chooses distributions that do not cover $[N]$ equally, we have to assume independence is constant across different supports (i.e. the assumption explained above).

**(ii) Different dimensions.** A more difficult setting of conflicting domains, is when we record different variables in each of the multi-origin datasets. In order for a DAG to be transportable, we *require* the variable sets to correspond. As such, we are only able to work with overlapping intersections of the non-overlapping domains. Doing so requires some additional assumptions on the noise: assuming we record some noise on each variable, we have to make the additional assumption that the noise is independent of the other variables, or at least the variables outside the intersection between domains. The latter is made quite often, and should not limit applicability of D-Struct in this setting too much (recall that applicability of D-Struct is mostly determined by the used DSF). The reason relates to the second assumption, below.

The second assumption is a bit stricter: any variables outside the intersection cannot be confounding variables inside the intersection. If two variables have no direct edges, and the nodes part of an indirect edge fall outside the domain-intersection, we have to expect the DSF to find an edge between these two nodes. While this direct edge is wrong, this is actually expected behaviour of most DSFs as the algorithms will find these variables to be correlated (due to the third, now unobserved, vari-

able). The only way to overcome these situations, is to use DSFs that naturally handle unobserved confounding.

**Related work.** Some work on structure discovery from multiple (non-overlapping) domains has been proposed. For example, Ghassami et al. [78] in the linear setting, Peters et al. [79] for the interventional setting, or Huang et al. [80] in the temporal setting. While the difference between the first ([78]) is clear (only focusing on linear systems, whereas we on a non-parametric setting), the others are not immediately clear. Some intuition into the difference can be achieved by considering that both the interventional and temporal *know* where the difference in distribution is coming from. So much so, that the known difference is exploited when garnering (causal) structural information. We believe applying our findings on transportability to the settings described earlier can be a promising new avenue of research.

## D  DEFINITIONS

**Definition 2** (Markov blanket.). *A Markov blanket of a random variable $X_i$ in a random set $\mathcal{X} := \{X_1, \ldots, X_d\}$ is any subset $\mathcal{X}' \subset \mathcal{X}$ where, when conditioned upon, results in independence between $\mathcal{X} \setminus \mathcal{X}'$ (the other variables) and $X_i$,*

$$X_i \perp\!\!\!\perp \mathcal{X} \setminus \mathcal{X}' | \mathcal{X}'. \tag{6}$$

*We will denote the Markov blanket of $X_i$ as $\mathcal{X}'(X_i)$.*

In principle, Def. 2 means that $\mathcal{X}'$ contains all the information present in $\mathcal{X}$ to infer $X_1$. Note that this does not mean that $\mathcal{X} \setminus \mathcal{X}'$ contains *no* information to infer $X_1$, but variables in $\mathcal{X}'$ are sufficient to predict $X_1$.

One step further, is a *Markov boundary* [53]:

**Definition 3** (Markov boundary.). *A Markov boundary of a random variable $X_i$ of a random set $\mathcal{X} := \{X_1, \ldots, X_d\}$ is any subset $\mathcal{X}^- \subset \mathcal{X}$ which is a Markov blanket (Def. 2) itself, but does not contain any proper subset which itself is a Markov blanket. We will denote the Markov boundary of $X_i$ as $\mathcal{X}^-(X_i)$.*

We can relate the Markov boundary (Def. 3) to probabilistic graphical modelling, as from a simplified factorisation (in eq. (1)), we can compose a Bayesian network. Specifically, each variable $X_j \in \mathcal{X}^-(X_i)$ depict one of three types of relationships: $X_j$ is a parent of $X_i$, denoted as $\mathrm{Pa}(X_i) = X_j$; $X_j$ is a child of $X_i$, denoted as $\mathrm{Ch}(X_i) = X_j$; or $X_j$ is a parent of a child of $X_i$, denoted as $\mathrm{Pa}(\mathrm{Ch}(X_i)) = X_j$. Assuming that $\mathbb{P}_\mathcal{X}$ is governed by a Markov random field (rather than a Bayesian network) simplifies things, as the Markov boundary depicts only directly connected variables.

While the above may suggest that the Markov boundary only implies a vague graphical structure, doing this for ever variable in $\mathcal{X}$ will strongly constraint the possible graphical structures respecting any found independence statements. D-separation (Def. 4) is then used to further limit the set of potential DAGs [28, 34]. Relating above definitions to those discussed in Section 2.1. For more information regarding the above, we refer to Koller and Friedman [26].

**Definition 4** (d-separation [34].). *In a DAG $\mathcal{G}$, a path between nodes $\mathcal{X}_i$ and $\mathcal{X}_j$ is blocked by a set $\mathcal{X}_d \subset \mathcal{X}$ (which excludes $\mathcal{X}_i$ and $\mathcal{X}_j$) whenever there is a node $\mathcal{X}_k$, such that one of two holds:*

*(1) $\mathcal{X}_k \in \mathcal{X}_d$ and*

$$\begin{aligned} &\mathcal{X}_{k-1} \leftarrow \mathcal{X}_k \leftarrow \mathcal{X}_{k+1}, \\ or \quad &\mathcal{X}_{k-1} \rightarrow \mathcal{X}_k \rightarrow \mathcal{X}_{k+1}, \\ or \quad &\mathcal{X}_{k-1} \leftarrow \mathcal{X}_k \rightarrow \mathcal{X}_{k+1}. \end{aligned}$$

*(2) neither $\mathcal{X}_k$ nor any of its descendants is in $\mathcal{X}_d$ and*

$$\mathcal{X}_{k-1} \rightarrow \mathcal{X}_k \leftarrow \mathcal{X}_{k+1}.$$

*Furthermore, in a DAG $\mathcal{G}$, we say that two disjoint subsets $\mathcal{A}$ and $\mathcal{B}$ are d-seperated by a third (also disjoint) subset $\mathcal{X}_d$ if every path between nodes in $\mathcal{A}$ and $\mathcal{B}$ is blocked by $\mathcal{X}_d$. We then write*

$$\mathcal{A} \perp\!\!\!\perp_\mathcal{G} \mathcal{B} | \mathcal{X}_d.$$

*When $\mathcal{X}_d$ d-seperates $\mathcal{A}$ and $\mathcal{B}$ in $\mathcal{G}$, we will denote this as d-sep$_\mathcal{G}(\mathcal{A}; \mathcal{B} | \mathcal{X}_d)$.*

**Definition 5** (Faithfulness from Peters et al. [34].). *Consider a distribution $\mathbb{P}_{\mathcal{X}}$ and a DAG $\mathcal{G}$*

*(i) $\mathbb{P}_{\mathcal{X}}$ is faithful to $\mathcal{G}$ if*

$$\mathcal{A} \perp\!\!\!\perp \mathcal{B}|\mathcal{C} \Rightarrow \mathcal{A} \perp\!\!\!\perp_{\mathcal{G}} \mathcal{B}|\mathcal{C},$$

*for all disjoint sets $\mathcal{A}, \mathcal{B}$ and $\mathcal{C}$.*

*(ii) a distribution satisfies causal minimality with respect to $\mathcal{G}$ if it is Markovian with respect to $\mathcal{G}$, but not to any proper subgraph of $\mathcal{G}$.*

Part (i) posits an implication that is the opposite of the global Markov condition

$$\mathcal{A} \perp\!\!\!\perp_{\mathcal{G}} \mathcal{B}|\mathcal{C} \Rightarrow \mathcal{A} \perp\!\!\!\perp \mathcal{B}|\mathcal{C},$$

for which we refer to Peters et al. [34, Def. 6.21].

Part (ii) is actually implied when part (i) is satisfied, when $\mathbb{P}_{\mathcal{X}}$ is Markovian w.r.t. $\mathcal{G}$, as per Peters et al. [34, prop. 6.35]. To have an idea for when faithfulness is not satisfied, we refer to Zhang and Spirtes [81] and Spirtes et al. [74, Theorem 3.2].

## E    INCORPORATING PRIOR KNOWLEDGE ON $\mathcal{I}(\mathbb{P})$ USING L-BFGS-B

Consider the following, where we wish to discover a structure between 3 variables: $X, Y, Z$, where the ground truth satisfies $X \perp\!\!\!\perp Y|Z$. According to the rules of $d$-speration (cfr. Def. 4), we are always in a structure where $X$ and $Y$ are *only* directly connected to $Z$, i.e. no direct connection between $X$ and $Y$ exists. Let us further assume that the system is linear (as this is what vanilla NOTEARS assumes, but without loss of generality towards recent NOTEARS extensions), then we have the following,

| structural equations | structure | adjacency matrix |
|---|---|---|
| $\begin{aligned} X &:= \epsilon_X, \\ Z &:= \beta_{Z,X}X + \epsilon_Z, \\ Y &:= \beta_{Y,Z}Z + \epsilon_Y, \end{aligned}$ | $(X) \longrightarrow (Z) \longrightarrow (Y)$ | $A = \begin{pmatrix} 0 & 0 & 1 \\ 0 & 0 & 0 \\ 0 & 1 & 0 \end{pmatrix}.$ |

Naturally, using only conditional independence, the direction of the arrows are not identifiable as explained above. However, NOTEARS is unable to narrow it down to the equivalence classes expressed in Def. 4. The reason is simple, NOTEARS' three optimisation components (the $h$-measure, an $L_2$ loss, and an $L_1$ regularizer on $A$, [76]) are satisfied exactly the same with the following system:

| structural equations | structure | adjacency matrix |
|---|---|---|
| $\begin{aligned} X &:= \epsilon_X, \\ Z &:= \beta_{Z,X}X + \epsilon_Z, \\ Y &:= \beta_{Y,X}X + \epsilon'_Y, \end{aligned}$ | 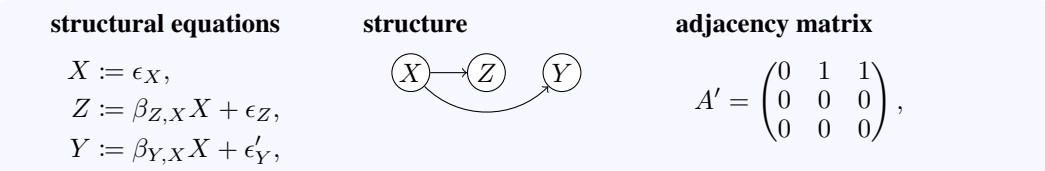 | $A' = \begin{pmatrix} 0 & 1 & 1 \\ 0 & 0 & 0 \\ 0 & 0 & 0 \end{pmatrix},$ |

where $\beta_{Y,X} = \beta_{Y,Z}\beta_{Z,X}$, and $\epsilon'_Y = \beta_{Y,Z}\epsilon_Z + \epsilon_Y$ resulting in $Y$ being determined again by a simple linear equation. Both systems allow the same data to be generated, however under the constraint that $X \perp\!\!\!\perp Y|Z$ only the former is possible.

We argue that NOTEARS (and extensions) are unable to differentiate between them. Consider the components optimised by NOTEARS: both solutions propose a DAG (i.e. $h(A) = h(A') = 0$); each DAG has an equal amount of arrows, leading to the same $L_1$-loss across $A$ and $A'$; and each equation is linear so NOTEARS is able to perfectly converge to each solution using its $L_2$ loss. Given that each component scores exactly the same, NOTEARS is unable to differentiate between these two results. Crucially however, in the latter system $X$ is *always* dependent of $Y$, resulting in $X \not\perp\!\!\!\perp Y|Z$ (and even $X \not\perp\!\!\!\perp Y$ eliminating v-structures) which is completely opposite to the former system.

**Prior Markov independencies.** We can however force known independence statements into DSFs a priori, using the L-BFGS-B optimizer. For example, consider the following $I = X_i \perp\!\!\!\perp X_j | Z$. If $I$ is known a priori, then we also know there cannot (under any circumstance) exist a direct link between $X_i$ and $X_j$ as this would immediately contradict $I$ which in turn would invalidate a structure proposing such a link.

As such, we propose to fix these directed edges to $0 \rightarrow \mathcal{A}_{ij}(\mathcal{G}), \mathcal{A}_{ji}(\mathcal{G})$, and exclude them from gradient calculation. This will not only constraint each DSL in step 2 above resulting in easier convergence, but it will also enforce any known $\mathcal{I}(\mathbb{P}_\mathcal{X})$ to be taken into account. Setting $A_{X,Y} = A_{Y,X} = 0$ would immediately restrict NOTEARS from converging to this false solution as the solution would require $A_{X,Y}$ to be 1. The same approach is currently used in NOTEARS (and consequentially D-Structs parallel DSFs), by setting bounds of each diagonal element in $A$ to $(0,0)$.

Setting some elements to $0$ using the L-BFGS-B bounds, we effectively limit the set of possible solutions. In fact, when applied to the above problems, the second solution would sit *outside* the set of possible solutions, ensuring that NOTEARS cannot converge to it.

# F    ADDITIONAL DETAILS ON SUBSAMPLING FROM DIFFERENT DISTRIBUTIONS

In Section 3.2 we introduced a method to sample subsets from a single-origin dataset such that the subsets correspond to distinct user-defined distributions. To provide some additional detail, we shall first discuss the general case, and then move on to discuss how we implemented this in D-Struct.

## F.1    THE GENERAL WAY

A high-level view on our subsampling routine is provided in Fig. 3. From Fig. 3 we learn that we need two ingredients for our subroutine to work:

1. We need a dataset that spans some domain $\mathcal{X}$. We can retrieve this domain simply by calculating the maximum and minimum value of each dimension in $\mathcal{X}$. *We have illustrated a simple dataset in Fig. 3a.*
2. We need a set of $K$ distinct distributions that span $\mathcal{X}$. In principle there is no constraint on these, besides them being different from one another, and each region in $\mathcal{X}$ having a non-zero probability of being sampled. *This is illustrated in Fig. 3b.*

Using the above two ingredients, we create $K$ empty subsets. For each subset, we then define one distribution, illustrated in Fig. 3b. In Fig. 3 we used a Gaussian for each subset as they span the domain, and are simple to evaluate. Using these distributions, we will fill each subset using data from Fig. 3a. Each data point in our dataset is evaluated $K$ times: using the user-defined distributions in Fig. 3b, we either include the sample in the corresponding subset, or not. When the probability of being sampled is *high enough*, it is included, when it is not high enough, it is excluded. High enough could be determined by something simple as a threshold, or something less parametric as a Bernoulli experiment. When finished, the subsamples look like Fig. 3c.

Alas, Gaussian distributions become more difficult to handle with increasing dimensionality as data is spread sparser in high dimensions. The provided high-level example may serve well as a (visual) explanation of our subroutine, it does not work well in practice. As such, we used a different implementation for D-Struct, which we explain in Section 3.2, and in more detail below.

## F.2    HOW IT'S IMPLEMENTED IN D-STRUCT

Recall that the main issue with the simple Gaussian implementation above is that it does not scale well to high-dimensions. As such, we need a different implementation that scales to high-dimensions.

**Defining the distributions.** We do this using a very simple idea: rather than sampling in covariate-space, we sample the dataset's *indices*, which correspond to a sample's covariates. However, before we do this, we need to make sure that the indices are in some way correlated with the covariates, which is not the case for a standard dataset as they are sampled i.i.d.

To provide some correlation between index and covariates, we first sort the covariates and reindex the dataset. This way, a smaller set of covariates now corresponds with a smaller index-value. Note that it is unimportant whether we sort descending or ascending, the only thing that matters is that there is some *logical* ordering.

Having an index that is correlated with the covariates allows us to define a distribution over the indices (which are one-dimensional) rather than over the covariates (which are $d$-dimensional). We chose the beta distribution as our user-specified distribution, where each of the $K$ distributions is given different parameters. The advantage a beta distribution has, is their flexibility to move its density over the entire domain (contrasting Gaussian distributions which are symmetrical). This point is illustrated in Fig. 4.

**Sampling data.** Once we have defined our distributions, we can use them to sample data. As with our high-level idea in Appendix F.1, we will evaluate each data point $K$ times to determine whether or not it should be included in each subset. However, rather than evaluating the chosen distributions using the covariates directly, we now use the index instead. Regardless of the number of dimensions we have, the index remains one-dimensional.

Evaluating a sample in D-Struct is done using a Bernoulli experiment: with the beta distributions we query the probability of being sampled and provide it to a Bernoulli experiment, the outcome determines inclusion or exclusion.

## G  CIT-BASED METHODS, SCORE-BASED METHODS AND FAITHFULNESS

### G.1  CIT-BASED METHODS

CIT-based methods such as the well known PC-algorithm, the SGS algorithm, or the inductive causation (IC) algorithm all require faithfulness as per Def. 5. The reason is such that they render the Markov equivalence class identifiable. As we have explained in Section 3.1, using d-separation we have a one-to-one correspondence to this class of DAGs. Any query of a d-separation statement can therefore be answered by checking the corresponding conditional independence test [15].

Most CIT-based methods have 2 main phases, based on a set of conditional independence statements. Assuming the latter is a correct set (that is, we have correctly inferred all the independence statements present in $\mathbb{P}_{\mathcal{X}}$ ), we first infer a skeleton graph, and then orient the edges. After these two phases, we have either a fully identified DAG, or a Markov equivalence graphs in case there are edges we were not able to orient.

**Phase 1: inferring a skeleton.** Based on lem. 1 (below) introduced in Verma and Pearl [75], the SGS and IC algorithm build a skeleton from a completely unconnected graph.

**Lemma 1.** *The following two statements hold:*

*(i) Two nodes $X$ and $Y$ in a DAG $(\mathcal{X}, \mathcal{E})$ are adjacent iff they cannot be d-separated by any subset $\mathcal{S} \subset \mathcal{X} \setminus \{X, Y\}$.*

*(ii) If two nodes $X$ and $Y$ in a DAG $(\mathcal{X}, \mathcal{E})$ are not adjacent, then they are d-separated by either $Pa_X$ or $Pa_Y$.*

Clearly, by using above lemma, SGS [74] and IC [28] *require* faithfulness. Contrasting methods that build from an unconnected graph, is the PC-algorithm which does the reverse: PC starts with a fully connected graph and step-by-step removes edges when they violate (ii) in lem. 1. While a different approach, both require d-separation, i.e. this too requires faithfulness to hold!

**Phase 2: orienting the edges.** As per Meek [39] there exists a set of graphical rules that is shown complete to correctly orient the edges based only on d-separation. Of course, this requires a *complete* set of correct independence statements which is arguably a much stricter assumption than faithfulness.

Essentially, we can relax the assumption of a complete set of independencies, but we'll have to replace it with other assumptions. One such example is assuming a $\mathbb{P}_{\mathcal{X}}$ to be Gaussian (which is also quite strict, but it serves our example). With the latter assumption, we can test for *partial correlation* [34, Appendices A.1 and A.2], which allows to identify the underlying Markov equivalence class

[82]. Furthermore, by additionally assuming a condition called *strong faithfulness* [83, 84], we have uniform consistency [82]. We refer to Peters et al. [34, Ex. 7.9] for an example.

## G.2 (DIFFERENTIABLE) SCORE-BASED METHODS

Contrasting CIT-based methods, are score-based methods. Score-based methods generalise our differentiable score based methods and non-differentiable methods. Contrasting CIT-based methods, which directly encode the independence statements governing $\mathbb{P}_{\mathcal{X}}$ into $\mathcal{G}$, a score-based method will evaluate $\mathcal{G}$ on how well it fits the observed data. The rationale behind these score-based methods is that wrongly encoded independence statements will yield poor model fits [85, 86].

We can formalise a score-based method as a function $S$ which is to be optimised over candidate DAGs:
$$\hat{\mathcal{G}} \coloneqq \underset{\mathcal{G} \text{ DAG over } \mathcal{D} \in \mathcal{X}}{\arg\max} S(\mathcal{D}, \mathcal{G}).$$

As such, there are two elements that comprise a score-based method: (i) the function $S$, and (ii) the way we optimise $S$. In our case, that is:

(i) $S$ corresponds to eq. (5), which is in large part determined by the underlying DSF through $\mathcal{L}_{\text{DSF}}$.

(ii) $S$ is optimised using gradient-optimisation, which has proven very efficient in this problem setting

Importantly, that rationale behind these methods does *not* require the faithfulness assumption for them to work. The latter may lead to violations against d-separation in case faithfulness does hold. However, in Appendix E we show how we can combat against this, by also incorporating any known independencies into our graph (which *does* require the faithfulness assumption to hold for those independence statements) using the L-BFGS-B optimisation algorithm.

