# OpenReview forum: "Differentiable and transportable structure learning"
_ICLR.cc/2023/Conference — Submitted to ICLR 2023_

### Official Review · Reviewer_nxCL · 2022-10-24

**Confidence:** 4
**Clarity, Quality, Novelty And Reproducibility:** Details are provided in the comments …
**Correctness:** 3
**Technical Novelty And Significance:** 2
**Empirical Novelty And Significance:** 2
**Recommendation:** 3

**Strength And Weaknesses:**

### Strengths
- The problem studied is highly relevant, since learning transportable structures may be an importance topic especially because of data heterogeneity across different datasets.
- The paper is well written and easy to follow.

### Weaknesses
- The setting studied is very similar to (and seems to be a special case of) differentiable federated structure learning [1, 2], which should be discussed. E.g., [1] considered data heterogeneity difference (i.e. different data distribution across multiple clients, but the underlying DAG is the same), which is essentially the same as transportability in this paper. If I understand correctly, the only key difference appears to be that [1, 2] additionally considered privacy issue.
- Therefore, the resulting approach is also similar to [1, 2]. Specifically, [1] compute the average adjacency matrix in each step and enforces different clients to use the same average matrix, while [2] enforces the adjacency of different clients to be the same via ADMM. The only key difference appears to be that the proposed approach uses a soft regularization scheme to *encourage* the same structure, while [1, 2] used a hard scheme to *enforce* the same structure.
- Following the comment above, although adding a soft regularization scheme encourages the same structure, it does not guarantee/enforce it, which I think might be a limitation (in practice the participating clients may want to learn a shared structure that is the same). This is also acknowledged by the experiment in Sec. 4.
- Adding a soft regularization directly over the parameters $A$ (which correspond to both structure and model parameters) may lead to the wrong structure. I.e., this regularization may conflict with the original least squares loss of the specific dataset when the data distributions are different. In my opinion, the backbone method Mcsl used by [1] might seem like a better choice (as compared to Notears-MLP) because it decomposes the structure and model parameters, and so the proposed regularization can be applied w.r.t. the structure parameters only.
- I did not manage to find experiments of different datasets having different data distributions in Sec. 4, which I think are much more relevant (considered by [1]), since that's the key motivation of using transportability (Definition 1) or invariance.
- The considered setting is highly similar to (invariant) causal discovery from multiple domains or heterogenous data, e.g., [3, 4, 5], which should be discussed and compared.

### Minor/Other Comments
- Sec 2.2: $\rho$ and $\lambda_2$ are not "hyperparamers", but rather the parameters of augmented Lagrangian method that will be updated during the optimization process.
- Sec. 2.2 "One can traverse G_X smartly to arrive at a DAG much faster": There was no guarantee that DSF arrives at a DAG in (Zheng et al, 2018); this was proved by [6, 7] and should probably be mentioned. Also, to the best of my knowledge, Zheng et al. (2018) did not claim/demonstrate that Notears runs "much faster" than CIT-based methods like PC, and I would encourage the authors to add a reference regarding this.
- For Notears-MLP, is the regularization scheme applied to the weighted adjacency matrix constructed w.r.t. the MLP weights, or all weights of the MLPs? Alg. 2 seems to indicate the latter, but the former sounds more intuitive to me.
- The efficiency gain in Fig. 5 is interesting and surprising. Is it because the for loop in fifth line of Alg. 1 can be run in parallel?

1. Federated causal discovery, 2022.
2. Towards Federated Bayesian Network Structure Learning with Continuous Optimization, 2022.
3. Multi-domain Causal Structure Learning in Linear Systems, 2018.
4. Causal inference using invariant prediction: identification and confidence intervals, 2015.
5. Causal Discovery from Heterogeneous/Nonstationary Data, 2020.
6. On the Convergence of Continuous Constrained Optimization for Structure Learning, 2022.
7. DAGs with No Fears: A Closer Look at Continuous Optimization for Learning Bayesian Networks, 2020.

**Summary Of The Paper:**

This paper develops an extension of Notears, a recent approach for structure learning with continuous optimization, to learn transportable structures (transportable in the sense that the learned structures are the same across multiple datasets with potentially different distributions). This is achieved by incorporating a regularization term in the optimization procedure to encourage the structure parameters over different dataset to be equal to one another. Empirical results are provided to validate the proposed method.

**Summary Of The Review:**

- A number of highly related works were not discussed and compared to. Specifically, the proposed method is similar to [1, 2].
- The proposed method adds a regularization term w.r.t. both structure and model parameters, which may lead to the wrong structure in practice especially when the data distributions are different for different datasets.
- The paper does not consider experiments of different data distributions for different datasets, which may be more common in practice and is the key motivation of transportability/invariance.

---

> ### Author Response · Authors · 2022-11-16
> **Thank you for your review (part 1)**
>
> _Dear reviewer nxCL, thank you for your detailed review of our paper. Below, we have provided a point-by-point response to your remarks. We look forward engaging further, if necessary._
>
>
> **Federated structure learning.** Thank you for sharing these works. We agree that they are relevant and have added discussion of these works in our revised paper. We provide a brief comparison here also.
>
> While we agree that the federated setting is related, there are a number of differences between the federated setting and the setting considered in our paper.
>
> First, we note that federated learning always assumes different clients, each with a different dataset. A major contribution in our paper, however, is the ability to leverage transportability even with a single dataset. None of the works in federated learning are able to do this.
>
> Furthermore, from [2]:
>
> > We assume here that the data from different clients follows the same distribution, [...]
>
> Which is different from our paper where we allow the distributions to be {\it different}; from our Definition 1:
>
> > [...] sampled from potentially different distributions [...]
>
> We hope the above makes it clear that the settings are different.
>
> Finally, you comment that transportability "seems to be a special case of differentiable federated structure learning [(FSL)]". Given the above, we would argue that transportability is more general, with FSL the special case of transportability.
>
> **_Changes to our manuscript:_** We apologise for not including these works in our paper -- we were not aware of them. We have now discussed and referred to these related works in the introduction of our paper.
>
> **Hard and soft regularization.** If there exists an underlying (true) structure, than each subset of the data should respect that structure. At least, this is what is (either implicitly or explicitly) assumed when trying to learn {\it one} structure. In that sense, while the reviewer is correct that our optimisation strategy ("soft" regularisation) is different from others ("hard" regularisation), we fail to see why ours would lead to wrong estimates. If our regulariser is minimised to zero, we show in Remark 1 (now Theorem 1) that the found adjacency matrices can only be exactly the same. Our regularisation strategy also keeps our entire architecture differentiable, which allows incorporation into methods using structure learners as, for example, auxiliary tasks such as [46].
>
> **Guarantees.** We refer here to our Theorem 1 (previously Remark 1) which states that eq. (4) is only zero if all of the DAGs are the same. If additionally the remainder of our loss in eq. (5) is 0, we benefit the same guarantees as the underlying structure learner. Naturally, this cannot be the case if the transportability assumption is violated (i.e. each subset respects a different DAG).
>
> **Parameters of NOTEARS-MLP.** Note that while a real matrix, the structural equations of the DAG found by NOTEARS-MLP are not fully specified by the adjacency matrix only as that would imply a linear model while NOTEARS-MLP (and Sobolev in our Appendix A.3) is non-parametric. Reducing the non-parametric structural equations to one real matrix is simply not possible.
>
> **Heterogeneous data from multiple domains.** We thank the reviewer for these additional references, now discussed in our paper. We further wish to point the reviewer to our Appendix C where we comment on transportability in non-overlapping domains.
>
> We have included a paragraph in our Appendix C to reflect this:
>
> > **Related work.** Some work on structure discovery from multiple (non-overlapping) domains has been proposed. For example, Ghassami et al. (2018) in the linear setting, Peters et al. (2016) for the interventional setting, or Huang et al. (2020) in the temporal setting. While the difference between the first ((Ghassami et al., 2018)) is clear (only focusing on linear systems, whereas we on a non-parametric setting), the others are not immediately clear. Some intuition into the difference can be achieved by considering that both the interventional and temporal know where the difference in distribution is coming from. So much so, that the known difference is exploited when garnering (causal) structural information. We believe applying our findings on transportability to the settings described earlier can be a promising new avenue of research.
>
>
> **_Changes to our manuscript:_** We have extended Appendix C to also include the references provided by the reviewer and updated our discussion of the theoretical properties of our regularization term.
>
> `continued in next comment...`

---

> > ### Author Response · Authors · 2022-11-16
> > **Thank you for your review (part 2)**
> >
> > **Minor comments.** Thank you for these helpful comments. We have now uploaded a version of our paper with clarifications as per your suggestions/questions. Below, we respond to each of these in term.
> >
> > **1. efficiency.** With respect to your comment on efficiency, you are correct that the original NOTEARS paper did not conduct an experiment w.r.t. computation time to be compared with more traditional methods. However, it is heavily suggested in the conclusion of their paper:
> >
> > > [...] By using second-order methods, each iteration make significantly more progress than first-order methods. Furthermore, although in practice not many iterations ($t \leftarrow 10$) are required, we have not established any worst-case iteration complexity results. In light of the results in Section 5.3, we expect there are exceptional cases where convergence is slow. Notwithstanding, NOTEARS already outperforms existing methods when the in-degree is large, which is known difficult spot for existing methods. We leave it to future work to study these cases in more depth
> >
> > **2. Regularisation.** With respect to your question:
> >
> > > For Notears-MLP, is the regularization scheme applied to the weighted adjacency matrix constructed w.r.t. the MLP weights, or all weights of the MLPs? Alg. 2 seems to indicate the latter, but the former sounds more intuitive to me.
> >
> > We do indeed update the adjacency matrix. However, we find this to be more intuitive when the object of interest is the adjacency matrix. We wish to stress here that our objective is _structure_, not the _structural equations._ After all, the structural equations may change across domains (i.e. they could be the reason for different distributions). What we assume to remain fixed is dependence and independence.
> >
> > **Fig. 5.** We did not run each structure learner in parallel. We believe the major reason for our performance increase is the amount of data the augmented Lagrangian of NOTEARS requires to be optimised over. With more data, optimisation takes longer. By splitting the data, we reduce the number of samples for each of the individual learners. Having more learners splits the data further, which seems to be beneficial, compared to having one learner optimising over the entire dataset.
> >
> > Of course, running the structure learners in parallel would increase computational performance even further!
> >
> > **_We would like to thank the reviewer for their detailed and instructive review. We hope our rebuttal has addressed their concerns. If this should not be the case, we would love to engage further!_**

---

### Official Review · Reviewer_auHu · 2022-10-25

**Confidence:** 4
**Clarity, Quality, Novelty And Reproducibility:** The problem statement is not very cle…
**Correctness:** 2
**Technical Novelty And Significance:** 1
**Empirical Novelty And Significance:** 1
**Recommendation:** 3

**Strength And Weaknesses:**

Weakness:

1. The problem setting is quite restricted.
2. There is no theoretic guarantee that the proposed algorithm will find a set of consistency DAG.
3. The experiment is quite weak. Instead of only using nonlinear models, the authors may also add some linear experiments, where the DAG is known to be identifiable.

**Summary Of The Paper:**

The authors consider the problem of learning DAG from heterogeneous datasets that share the same DAG. This problem statement is quite restricted. From Eq(4) we can see that the authors not only encourage the DAG to have the same structure, but also to have the same parameter. While in Figure 1, the authors assume that multiple datasets to have different distribution. In this case, there is only one possibility, that is the distribution of noise variable may be different. In the NOTEARS framework, the only possibility is that the Gaussian noise may have different mean and variance in different datasets. This setting is in fact quite restricted.

**Summary Of The Review:**

The paper proposed a method to learn DAG from heterogeneous datasets, but the setting in the paper is quite restricted and the experiments are also weak.

---

> ### Author Response · Authors · 2022-11-16
> **Thank you for your review**
>
> _Dear reviewer auHu, thank you for reviewing our paper. Below, we have provided a point-by-point response to your remarks. We hope this addresses all of your concerns, and look forward engaging further with you._
>
> **Problem setting.** Requiring multiple datasets, with different distributions, that span the same domain is actually a rather common setting (and in medicine, quite a frequent one). Imagine a company that operates in different areas (such as e-commerce), collecting the same variables. Or a government of a country with multiple provinces. However, as stated in the introduction our paper:
>
> > [...] our method is not limited to [settings with multiple datasets] alone. In fact, we find that enforcing transportability significantly increases performance in settings with one dataset, which is arguably most common.
>
> Learning a structure from a dataset ($\mathcal{D} \to \mathcal{G}$) is a very general setting. In fact, one could apply our method to _any_ tabular dataset. We would agree with the reviewer if D-Struct is only applicable in the multi-domain setting, but this is not the case.
>
> Furthermore, you also make the comment that "the Gaussian noise may have different mean and variance in different datasets". This is quite a general setting. Other changes in the distribution would result in different DAGs for each domain, which would make learning over multiple domains a useless exercise. Assuming shared structure is indeed quite general and the sole reason for pooling data in most settings.
>
> As the above comment is aimed mainly towards using NOTEARS as an underlying structure learner, we wish to emphasise: (i) that D-Struct is not restricted to NOTEARS, we chose to use NOTEARS because (ii) it is so widely adopted. Far from being restricted, NOTEARS has been applied to very diverse areas. Our work is applicable to the same settings as NOTEARS.
>
> **Theoretical guarantee.** We refer to Theorem 1 (previously named Remark 1), which provides a guarantee on our regularization term. In this theorem, we prove that the only way eq. (4) can be zero is when all the adjacency matrices are the same (i.e. a consistent DAG). Reducing eq. (4) to 0, only leaves $\mathcal{L}_{\text{DSF}}$ in eq. (5), which inherits the same guarantees as the underlying structure learning. In the case of NOTEARS we refer to [1, 2].
>
> **Experimental setup.** Please note that our experiments are the same as in NOTEARS-MLP (and Sobolev). We find our results compelling as they outperform NOTEARS in {\it their} experimental setup. Furthermore, we also added to the NOTEARS experiments to further explore transportability and subsampling.
>
> Note also that the linear setting is only identifiable under some assumptions, it is certainly not always identifiable [3].
>
> _We hope the above clears up some of the comments made in your review. If this is not the case, please, do not hesitate to ask for further clarifications. We would be happy to engage further._
>
> ### Additional references
>
> [1] On the Convergence of Continuous Constrained Optimization for Structure Learning, 2022.
>
> [2] DAGs with No Fears: A Closer Look at Continuous Optimization for Learning Bayesian Networks, 2020.
>
> [3] Causal Discovery with Continuous Additive Noise Models. Peters et al. JMLR (2014)

---

### Official Review · Reviewer_pcus · 2022-10-27

**Confidence:** 4
**Correctness:** 3
**Technical Novelty And Significance:** 2
**Empirical Novelty And Significance:** 3
**Recommendation:** 5

**Clarity, Quality, Novelty And Reproducibility:**

The writing is a bit sloppy in some places.

* The objective in equation (2) is many times referred to as the "score", which can be misleading since the score in a score-based method is simply what the authors write as $F(A)$ or the "regularized" score $F(A) + \lambda_1 * ||A||_1$ algorithm. The other terms w.r.t. $h$ stem from the augmented Lagrangian to solve the constrained problem.

* In Section 3.1., the authors also mention $K$ "distinct" DSFs. I believe this *cannot* be the case. For instance, consider linear SEMs with equal variances, from Loh & Buhlmann (2014) we know that the ground-truth DAG is the global minimizer of the least squares (LS) score, i.e., these are identifiable models. Now, suppose one uses LS for dataset D1 and another score, call this S2, for another dataset D2. Moreover, suppose we are able to find the global minimizes in each case. Then, using LS on D1 already finds the correct DAG, while using S2 on D2 will find an incorrect DAG. Forcing both DAGs to be the same, as in Dstruct, will fail to find the correct DAG. Thus, to me, it does not make sense to use "distinct" DSFs.

* Algorithms 1 and 2 are a bit confusing, Alg 1 does not use h_tol and Alg 2 does not have rho_max as input. Please polish these minor details.

* In the paragraph on transportability on Page 8, what do "internal" graphs mean?

As per novelty, I like the idea of trying to find a transportable DAG, but I am unclear if the approach really does make sense due to my questions above. I am under the impression that really the core technical contribution is simply adding an average of DAGs as a measure to force the different DAGs to be the same, which looks like a fine contribution but perhaps not enough to grant publication at this time.

I don't have many concerns about quality and reproducibility.





**Strength And Weaknesses:**

### Strengths

The core ideas of the paper are well-written and well-motivated. The authors do a good job of stating the problem under study and the contributions of their work. The method is simple to grasp and simple to adapt to existing differentiable approaches to learning DAGs.

### Weaknesses

The main weakness is the experimental section.

* Lack of proper comparison to other methods. First, in my opinion, the correct baseline should not be plain NOTEARS but the version of NOTEARS with the same exact sampling mechanism for Dstruct and taking the average or median as the final output, i.e., the only difference to Dstruct should be the L_MSE step. I stress this point because we need to make sure that (part of) the improvements are not due to the ensemble mechanism (which would be a very incremental contribution) but to the addition of the L_MSE regularization. Second, there really should be comparisons to other methods, especially Bayesian methods such as DiBS [1]. This is because in those methods we can sample DAGs from a posterior distribution and, again, perhaps taking the mode or mean would result in a transportable DAG. For these Bayesian methods, I would feed all the data---even from different distributions---and see if they are robust to these changes in distributions.

* The speed-up claims are rather disappointing, the graphs have a very small number of variables, $d=5$. To make a more precise claim about computational speed-ups there must be experiments for different values of d and not just n. This is because increasing $n$ really just makes the training of the neural nets longer but does not increase the complexity of enforcing acyclicity, for which $d$ does. Recent work [2] for instance ran the NOTEARS-MLP for up to 100 nodes leveraging a new acyclicity function.

[1] Lorch et al. (2021), "DiBS: Differentiable Bayesian Structure Learning"
[2] Bello et al. (2022), “DAGMA: Learning DAGs via M-matrices and a Log-Determinant Acyclicity Characterization”


**Summary Of The Paper:**

The paper studies the problem of learning DAGs from observational data. In contrast to previous work under the continuous framework, the authors aim to learn a DAG that is transportable to other distributions over the same observables that respect a set of conditional independences. In a few words, the approach consists of learning K different DAGs where each DAG is penalized by how far away it is from the mean of the K DAGs, in this way, the approach remains fully differentiable and the resulting DAGs are somehow forced to be similar or equal for the K different datasets. The authors apply a preprocessing sampling step to make the method work even for the case of observing a single dataset. Some experiments are provided to demonstrate the efficacy of the proposed method against the nonparametric NOTEARS algorithm.

**Summary Of The Review:**

My main concerns are listed above which currently make me inclined toward rejection.

---

> ### Author Response · Authors · 2022-11-16
> **Thank you for your review (part 1)**
>
> _Dear reviewer pcus, thank you for reviewing our paper. Below, we have provided a point-by-point response to your remarks. We hope this addresses all of your concerns, and look forward engaging further with you._
>
> **Baseline comparisons.** The reviewer suggests the following benchmark:
>
> > the correct baseline should not be plain NOTEARS but the version of NOTEARS with the same exact sampling mechanism for D-Struct and taking the average or median as the final output, i.e., the only difference to D-Struct should be the $\mathcal{L}_\text{MSE}$ step.
>
> We want to point the reviewer to our Appendix A.6, where we _did_ compare our method with NOTEARS using the same sampling mechanism, i.e. the only difference being to not include the $\mathcal{L}_\text{MSE}$ step. We have included these results below for the reviewer's convenience.
>
> [Fig 11](https://imgur.com/a/ZuelTMp)
>
> [Fig 12](https://imgur.com/a/5AUS4eM)
>
> The above two figures show that when doing everything the same, but eq. (4), the average structure is simply not a DAG. Imagine two conflicting DAGs (as the two figures show), where the conflict is simply an arrow that is pointed the opposite direction. Averaging these conflicting structures would no longer result in a DAG. Furthermore, it is eq. (4) that is making sure D-Struct is transportable, which is a property we wish to optimise for in our proposal.
>
>
> With respect to additional structure learners. We appreciate the need for additional methods (and refer to our Appendix A for exactly this), however we fail to see why they need to be Bayesian in particular. The reviewer states that:
>
> > This is because in those methods we can sample DAGs from a posterior distribution and, again, perhaps taking the mode or mean would result in a transportable DAG.
>
> Reducing the posterior to a Monte Carlo mean (or mode) is in essence no different from point-estimates provided by, for example, NOTEARS (as they estimate the expectation of a DAG). We stress that our setup is to provide different data segments to different methods. Doing so may result in different DAGs across these methods. Giving all the data to one learner will yield one DAG only, even if learned by a Bayesian method. To DiBS in particular: we note that DiBS actually enforces acyclicity by using the NOTEARS constraint (eq. (3) in our paper).
>
> To prove our point above, we have ran an additional experiment using DiBS. We ran DiBS three times on the same data we used for our experiments to validate D-Struct. From our experiments (see results below) we learn that DiBS is indeed not transportable: we get different DAGs. When combining the DAGs (by averaging), we find that the resulting DAG is indeed far removed from the ground-truth DAG.
>
>
> [DiBS experiment 1](https://imgur.com/a/rt8jY60)
>
> [DiBS experiment 2](https://imgur.com/a/rWCmEoN)
>
> **Increasing $n$ to measure speedup.** We thank the reviewer for this suggestion and have included the results on computation time with higher dimensional data. As seen in the below table, the conclusion remains the same - although the speedup reduces in magnitude, D-Struct remains more than twice as fast in all cases.
>
> | Parameter | D-Struct Speedup     |
> |-----------|--------------------- |
> | n (d=5)        |                      |
> | 200       | 13.79                |
> | 500       | 20.44                |
> | 1000      | 10.59                |
> | 2000      | 11.03                |
> | d (n=1000)|                      |
> | 5         | 10.59                |
> | 7         | 7.34                 |
> | 10        | 2.33                 |
>
> `continued in next comment...`

---

> > ### Author Response · Authors · 2022-11-16
> > **Thank you for your review (part 2)**
> >
> >
> > **Distinct DAGs.** We will rebuttal the following comment the reviewer makes with three different arguments:
> >
> > > In Section 3.1., the authors also mention $K$ "distinct" DSFs. I believe this cannot be the case. For instance, consider linear SEMs with equal variances, from Loh & Buhlmann (2014) we know that the ground-truth DAG is the global minimizer of the least squares (LS) score, i.e., these are identifiable models. Now, suppose one uses LS for dataset D1 and another score, call this S2, for another dataset D2. Moreover, suppose we are able to find the global minimizes in each case. Then, using LS on D1 already finds the correct DAG, while using S2 on D2 will find an incorrect DAG. Forcing both DAGs to be the same, as in Dstruct, will fail to find the correct DAG. Thus, to me, it does not make sense to use "distinct" DSFs.
> >
> > 1. We operate under the assumption that we _don't_ have access to correct model specification. The reviewer's argument is based on mixing an incorrectly specified model with a correctly specified model. The reviewer is correct, of course, but there is no way of knowing this information prior to learning the model. For example, learning a linear model will yield the correct DAG, only if the system is linear.
> >
> > 2. Furthermore, if one of the methods is indeed correctly specified, then performance on D2 _increases_ also. In essence, combining different DSFs will increase performance in general.
> >
> > 3. Finally, we also have an experiment (Figs 11 and 12 above) in our Appendix A, where we compare the different structures found by NOTEARS-MLP. In this setting, the DSF is actually correctly specified, yet still finds conflicting structures! In the case of your specific example, perfect identifiability is achievable, under the very strict assumption that the system is linear.
> >
> > The above answers directly respond to your comment. However, we would like to stress further that our setting (that of conflicting distributions) is more general than the one you describe. It is entirely possible that the linear systems may vary across D1 and D2 (for example a piece-wise linear system), yet still retain the same independence relations. Learning the structure using a transportable method (such as D-Struct), will find the corresponding structure, _despite_ having different coefficients. From this, it should be clear that we focus on structure only, not structural equations.
> >
> > **Novelty.** Your comment w.r.t. novelty is based on the comments you have (which we responded to) above:
> >
> > > As per novelty, I like the idea of trying to find a transportable DAG, but I am unclear if the approach really does make sense due to my questions above. I am under the impression that really the core technical contribution is simply adding an average of DAGs as a measure to force the different DAGs to be the same, which looks like a fine contribution but perhaps not enough to grant publication at this time.
> >
> > While adding the average regulariser (eq. (4)) is indeed a contribution in our paper (which to our knowledge has not been proposed previously), we have two additional contributions in our paper:
> >
> > * We propose the concept of Transportability, which we believe to be an important property of traditional structure learners, lacking from differentiable learners.
> >
> > * We also proposed a subsampling routine, which we show to be a necessary step in D-Struct as per Tab. 2 (and Tab. 6 in Appendix A.4).
> >
> > Note further that D-Struct improves an existing, widely-used structure learner in all aspects.
> >
> > **_We wish to end by thanking the reviewer once more for their constructive review. We hope the above addresses your comments. If there remain questions, please, do not hesitate to ask. We look forward to engaging further!_**

---

### Official Review · Reviewer_CF11 · 2022-10-31

**Confidence:** 3
**Correctness:** 4
**Technical Novelty And Significance:** 2
**Empirical Novelty And Significance:** 2
**Recommendation:** 5

**Clarity, Quality, Novelty And Reproducibility:**

clarity: good

quality: good

originality: problem is new, but technically it is limited.

**Strength And Weaknesses:**

Strength:
It seems this is a first work to learn one single DAG from multiple domain.
Subset construction is interesting, as it may improve the statistical property of the estimation.

Weakness:
The technical contribution is rather limited, with a MSE terms on graphs. It would be interesting for authors to discuss the choice of such a regularization against potential other alternatives.
Only one baseline is compared. Understandably this may be first work, but other works could and should be adapted as a baseline. For example, the multi-task DAG learning from [62].

Comments:
- subset construction: it is not fully clear how much the performance gain is from d-struct formulation, compared against an ensembled approach on different subset of data (maybe taking an average to obtain the final result with measure to ensure acyclicity).  This could also serve as a baseline.
- While transportability definition is clear, authors did not discuss much about it with respect to other standard problem regimes. For example, how is transportability problem different from a typical multi-task learning or transfer learning for DAGs?
- another way to improve technical contribution is to study feature differences or absences in different domains, and how to integrate them into one single DAG.

**Summary Of The Paper:**

The paper proposes transportability in DAG structure learning problem, which can be seen as a multi-task DAG learning problem or a cross validation method from a single task. The changes from NOTEARS methods is the addition from an average graph loss term additional to the differential scoring function. The method is compared with one baseline NOTEARS to show its superior performance in accuracy.

**Summary Of The Review:**

the paper addressed a new problem in DAG learning but with straightforward technical contribution.

---

> ### Author Response · Authors · 2022-11-16
> **Thank you for your review**
>
> _Dear reviewer CF11, thank you for your review. We have addressed the comments you have provided on a point-by-point basis below._
>
> **Benefits of subset construction.** To truly estimate the benefits of our subsampling routine, the reviewer suggests the following:
>
> > it is not fully clear how much the performance gain is from d-struct formulation, compared against an ensembled approach on different subset of data (maybe taking an average to obtain the final result with measure to ensure acyclicity). This could also serve as a baseline.
>
> Randomly sampling subsets (as opposed of using our subsampling routine) and executing D-Struct as intended should affect performance (if not, our subsampling has no use). The difference in performance with using D-Struct while also using our proposed subsampling routine would inform us about the benefits of using it.
>
> _We agree completely!_ In fact, we had already included an experiment that does exactly this. Consider the table below (Table 2 in our submission) where we report the SHD (lower is better) with and without subsampling. From Table 2, we learn that using our subsampling routine does indeed increase performance! For the full table on this experiment we refer to Appendix A.4 (Table 6).
>
>
> | _subsampling used_ | yes | no |
> | --- | --- | --- |
> | K | ||
> | 2 | **2.80** (0.53) | 3.40 (0.58) |
> | 3 | **3.00** (0.37) | 4.00 (0.59) |
> | 5 | **2.80** (0.57) | 4.40 (1.29) |
>
>
>
> **Multi-task learning, transfer learning, and transportability.** Let us first discuss multi-task learning. From the below quote of our paper (end of Sec. 2):
>
> > [...] Rather than trying to recover one true DAG, [a multi-task DAG learner's objective] is to learn multiple (potentially different DAGs) through a multi-task objective. In our work, we assume there does exist one (unique) DAG, which is the one we aspire to recover.
>
> We hope the above makes the difference between multi-task learning and transportability clear: each task in multi-task learning is different, whereas in transportability they are the same.
>
> Regarding transfer learning, the difference is more subtle, but there still is a difference! Transportability is a _property_ of a method, while transfer learning is an _objective_ of a method. Essentially, when doing transfer learning, we wish to learn a method on one dataset, and later adjust it such that it performs well on another. While a transportable method should maintain performance on both.
>
> However, there is some nuance here as well. Imagine the objective is to recover the true _causal_ DAG (which we do not claim to find using D-Struct). If we find it by learning on a first dataset, it should be obvious that it transfers to another (otherwise it is not the true causal DAG). The latter reminds us of transfer learning. Of course, if a method is truly able to find this DAG, it is also transportable. If the goal is to recover the causal DAG, we believe transfer learning and transportability aligns.
>
> **Non-overlapping domains.** The reviewer is correct that finding DAGs across domains is another area of interest in DAG-learning. However, it is not the one our paper operates in. We find it best to leave the adaptation of D-Struct in this new area a topic of future research. However, we would like to point the reviewer to our Appendix C where we _do_ comment on transportability in non-overlapping domains.
>
> **_Thanks again for reviewing our paper. We hope the above responses to your comments are satisfactory. Should there remain any questions or comments, do not hesitate to ask. We look forward to engaging further!_**

---

### Decision · Program_Chairs · 2023-01-20

**Decision:**

Reject

**Justification For Why Not Higher Score:**

The paper seems to have serious problems in many dimensions, as mentioned in the discussion above.

**Justification For Why Not Lower Score:**

N/A

**Metareview: Summary, Strengths And Weaknesses:**

The authors consider the problem of learning DAG from heterogeneous datasets that share the same DAG, called transportability, in the causal inference literature.

Overall, the problem is real and significant, and it's commendable the authors attempted to solve such a difficult task. Also, I appreciate the goal of finding a differentiable approach to the problem, which is usually solved through non-neural approaches. On the other hand, reviewers found the particular setting restricted, the method with no theoretical guarantees and/or incremental, and the experiments somewhat weak. The reply was appreciated but was not strong enough to revert some of the reviewers' concerns.

Since the work is quite interesting, they should try to improve the writing, taking into account all the feedback provided. Furthermore, the data structure that is the target of the learning process in this paper is an object already understood in the literature under the name of selection diagram (Pearl & Bareinboim,  (2011). As I understand, the study of transportability has been introduced in various papers by them (e.g.,  2013, 2016), and readers familiar with this literature may wonder whether the constraints implied by the causal system (formally, the selection diagram) in the data are encoded and leveraged by the method proposed in the paper. Is the proposed method complete (necessary and sufficient) relative to these constraints? Subsumed? More general? Making this connection could shed some light on the theoretical properties of the proposed work. Finally, other works have leveraged invariances across environments and proved their completeness (Jaber et al., NeurIPS, 2020) in some IT sense,  and it would be important to understand how these methods differ.